# Corrector Sampling in Language Models

**Itai Gat**      **Neta Shaul**      **Uriel Singer**      **Yaron Lipman**

FAIR, Meta

## Abstract

Autoregressive language models accumulate errors due to their fixed, irrevocable left-to-right token generation. To address this, we propose a new sampling method called Resample-Previous-Tokens (RPT). RPT mitigates error accumulation by iteratively revisiting and potentially replacing tokens in a window of previously generated text. Fine-tuning a pretrained 8B parameter model with RPT for only 100B resulted in ∼10% relative improvements on reasoning and coding benchmarks compared to the standard sampling.

## 1   Introduction

Autoregressive (AR) language models represent a pivotal advancement in sequence generation, delivering state-of-the-art results in translation, code generation, text summarization, question answering, and many other text tasks. By factoring the probability $p(x)$ of a sequence $x$ with the chain rule, AR training reduces to learning the next-token-prediction (NTP) conditional probabilities, making both training and inference highly efficient. At inference time, however, this left-to-right process is irrevocable: once a token is drawn the model cannot revise it, resulting in error propagating from the NTP sampling.

Recent progress in LLMs has largely focused on improving training via: (i) better data quality (Meta, 2024; DeepSeek, 2024; Groeneveld et al., 2024; Gemini, 2025; Li et al., 2025; OLMo, 2025; Lambert et al., 2025); (ii) modifications to the transformer architecture, such as, positional embeddings (Su et al., 2021; Press et al., 2022), attention mechanism (Ainslie et al., 2023; Beltagy et al., 2020); and (iii) Reinforcement learning fine-tuning (Ouyang et al., 2022; Ramesh et al., 2024; DeepSeek, 2025; Muennighoff et al., 2025; Kimi, 2025). Nevertheless, the *sampling* itself remains relatively underexplored, with most models still relying on the vanilla NTP sampling (Holtzman et al., 2020).

In this work, we introduce *Resample-Previous-Tokens* (RPT), a novel sampling method that iteratively revisits a fixed-size window of previously sampled tokens. Unlike NTP sampled tokens, which are unchangeable once sampled, RPT allows local token replacements during generation, potentially reducing error accumulation. RPT training is lightweight, allows fine-tuning pretrained AR models, preserves their NTP sampling quality and speed (i.e., with key-value caching), and integrates seamlessly into existing AR models and code. We fine-tuned a pretrained 8B AR model for 10% of its final training iterations and compared it to the fully-trained pretrained model to find RPT pretraining and sampling provides 5%-10% relative improvements in common benchmarks. Our contributions include: (1) Introducing RPT, a simple yet effective sampling process from LLMs; (2) Developing a training algorithm that incorporates seamlessly in AR training loop with minimal overhead and parameter cost; (3) Demonstrating empirically that RPT outperforms standard NTP sampling in both reasoning and coding tasks, as well as in a controlled error analysis.

39th Conference on Neural Information Processing Systems (NeurIPS 2025).

## 2 Resample-Previous-Tokens sampling

We denote by $x = (x_1, x_2, \ldots, x_n)$ a sequence of tokens, where each token $x_i$ is an element of a vocabulary set $\mathcal{V}$, i.e., $x_i \in \mathcal{V}$, and $V = |\mathcal{V}|$ denotes the size of the vocabulary. Autoregressive (AR) modeling uses the probability chain-rule to model the joint probability of sequences $x$,

$$p(x) = \prod_{i=1}^{n} p(x_i|x_{<i}), \tag{1}$$

where we use the notation $x_{<i} = (x_1, \ldots, x_{i-1})$, and learn a model (denoted with $\hat{*}$)

$$\hat{p}(x_i|x_{<i}) \approx p(x_i|x_{<i}) \tag{2}$$

that, in inference time, can be used for sampling from the joint $x \sim p$ via next-token-prediction (NTP) sampling:

$$\textbf{Next Token Prediction} \qquad x_i \sim \hat{p}(x_i|x_{<i}) \tag{3}$$

for $i = 1, 2, \ldots, n$. This procedure obviously introduces some errors into the NTP sampling process that originates from the approximation errors in equation 2.

Our goal is to reduce the errors of the NTP sampling process by using *Resample-Previous-Tokens* (RPT) sampling. In its simplest form, we consider pairs of adjacent tokens $x_i, x_{i+1}$, initialized with NTP sampling, and perform iterations of the form

$$\textbf{Resample Previous Tokens} \quad \begin{cases} x_i \sim \hat{p}(x_i \quad |x_{<i}, x_{i+1}) \\ x_{i+1} \sim \hat{p}(x_{i+1}|x_{<i+1}) \end{cases} \tag{4}$$

where the number of iterations is a hyper-parameter or set by threshold (e.g., confidence), see Figure 1 for an illustration. To implement RPT sampling we require, in addition to the learned NTP conditional $\hat{p}(x_i|x_{<i})$ (used in the second equation in 4), to learn the previous-token-prediction (PTP) conditional (used in the first equation in 4), i.e.,

$$\hat{p}(x_i|x_{<i}, x_{i+1}) \approx p(x_i|x_{<i}, x_{i+1}) \tag{5}$$

that predicts the $i$-th token $x_i$ given the previous tokens $x_{<i}$ *and* a future token $x_{i+1}$. A key observation of this paper is that since the error of predicting $x_i$ using a future token is smaller than standard NTP error, the overall error in the RPT sampling (4) can be shown (under certain conditions) to be smaller than NTP and often empirically leading to a better (yet more expen-

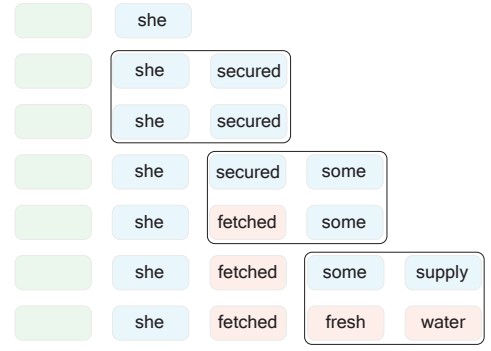

Figure 1: Windows (in black) show 2 iterations of RPT sampling (see equation 4); green is context $x_{<i}$; blue are tokens introduced with NTP and red are tokens changed during RPT iterations.

sive) sampling procedure than NTP. We will theoretically compare and analyze the errors in NTP and RPT sampling in the next section.

More generally, we can consider longer sequences of $w \in \mathbb{N}$ tokens, $x_i, \ldots, x_{i+w-1}$, where $w$ is called *window size*. Initialize it with NTP, and iterate

$$\begin{cases} x_i & \sim \hat{p}(x_i \quad |x_{<i+w}, \overline{x_i}) \\ x_{i+1} & \sim \hat{p}(x_{i+1} \ |x_{<i+w}, \overline{x_{i+1}}) \\ x_{i+2} & \sim \hat{p}(x_{i+2} \ |x_{<i+w}, \overline{x_{i+2}}) \\ \quad \vdots \\ x_{i+w-1} \sim \hat{p}(x_{i+w-1}|x_{<i+w-1}) \end{cases} \tag{6}$$

| Output | $p(x_2|x_{<2})$ | $p(x_3|x_{<3})$ | $p(x_3|x_4,x_{<3})$ | $p(x_5|x_{<5})$ | $p(x_6|x_{<6})$ | $\cdots$ | $p(x_n|x_{<n})$ |
|---|---|---|---|---|---|---|---|
| | Model | | | | | | |
| Input | $x_1$ | $x_2$ | $x_4$ | $x_3$ | $x_5$ | $\cdots$ | $x_{n-1}$ |

Figure 2: RPT training with window size 2: with some probability $q$ two adjacent tokens are swapped (bottom row, $x_3 \leftrightarrow x_4$), the model predicts the conditional probabilities as shown in the top row.

where $\overline{x}_i$ means that the token $x_i$ is *excluded* from the condition, that is,

$$(w_{<i+w}, \overline{x_i}) = (\ldots, x_{i-1}, x_{i+1}, \ldots, x_{i+w-1}). \tag{7}$$

Rearranging the indices, we need to learn conditionals of the form

$$\hat{p}(x_{i-\ell}|x_{<i+1}, \overline{x_{i-\ell}}) \approx p(x_{i-\ell}|x_{<i+1}, \overline{x_{i-\ell}}), \quad 0 \le \ell \le w - 1. \tag{8}$$

The $\ell = 0$ case corresponds to the NTP conditional. As above, all the conditional with $0 < \ell < w$, enjoy lower error than the standard NTP prediction, and in fact, the more future tokens are used (i.e., the larger $w - \ell$), the lower the error (see Figure 5). Lastly, we note that performing RPT with window size $w = 0$ collapse to standard NTP sampling.

## 2.1 Learning the conditionals required for RPT

For RPT sampling we need to learn to sample from conditionals of the form $p(x_{i-\ell}|x_{<i+1}, \overline{x_{i-\ell}})$, $0 \le \ell \le w - 1$ (see equation 8). To this end, we consider the following two sequences of indices: first, $\sigma = (\sigma_1, \ldots, \sigma_n)$, a permutation of $\{1, \ldots, n\}$ that encodes the order in which tokens are fed into the model, i.e., the $i$-th token that is fed into the model is $x_{\sigma_i}$; we denote the entire permuted sequence by $x_\sigma = (x_{\sigma_1}, x_{\sigma_2}, \ldots, x_{\sigma_n})$. We consider only a particular type of permutations, namely, where a random index $k \in \{1, \ldots, n\}$ is pushed $w - 1$ places to the right, that is

$$\sigma = (1, 2, \ldots, k-1, \overbrace{k+1, \ldots, k+w-1, \boldsymbol{k}}^{k \text{ moves } w-1 \text{ places to the right}} \quad , k+w \quad , \ldots, n). \tag{9}$$

The second sequence $\tau = (\tau_1, \ldots, \tau_n)$ prescribes the *target token index* $x_{\tau_i}$ for the model $i$-th output,

$$\tau_i = \begin{cases} i+1 & \text{if } \{\sigma_1, \sigma_2, \ldots, \sigma_i\} = \{1, 2, \ldots, i\} \\ k & \text{o/w} \end{cases}. \tag{10}$$

That is, for the permutation in equation 9 we have

$$\tau = (2, 3, \ldots, k \quad , \overbrace{\boldsymbol{k} \quad , \ldots, \boldsymbol{k}}^{\text{predict past } k\text{-th token}} \quad , k+w \quad , k+w+1, \ldots, n). \tag{11}$$

Note that all the tokens predict the next token, as encoded in the respective $\tau_i$, except those that correspond to the part marked by over-brace that specifically predict the (past) $k$-th token, $x_k$. With $\sigma, \tau$ our network models

$$\hat{p}(x_{\tau_i}|x_{\sigma_{<i+1}}) = \begin{cases} \hat{p}(x_{i+1}|x_{<i+1}) & \text{if } i < k \text{ or } i \ge k+w-1 \\ \hat{p}(x_{i-\ell}|x_{<i+1}, \overline{x_{i-\ell}}) & \text{o/w } 0 \le \ell = i - k < w - 1 \end{cases} \tag{12}$$

where we denote $x_{\sigma_{<i+1}} = (x_{\sigma_1}, x_{\sigma_2}, \ldots, x_{\sigma_i})$. The first case in equation 12 corresponds to standard NTP, while the second case corresponds exactly to the PTP conditionals required for our RPT sampling with window equal or smaller than $w$, as summarized in equation 8. Figure 2 shows an example corresponding to the choice of $\sigma = (1, 2, \boldsymbol{4}, \boldsymbol{3}, 5, 6, \ldots, n)$.

**Training** During training, we permute each given training data sequence with a probability of $s$ using a permutation $\sigma$ as defined in equation 9. The permutation is performed with $k$ sampled uniformly from the set $\{1, \ldots, n - w + 1\}$. In fact, due to the large sequence length typically used in text modeling, we perform the swap on more than a single token. In detail, we set some probability $q \in (0, 1)$ and then construct a permutation $\sigma$ as follows: (i) start with the identity permutation $\sigma = (1, 2, \ldots, n)$ and traverse from left to right with indices $k = 1, 2, \ldots$; (ii) for each index $k$, with probability $q$ we move it forward by $w - 1$ places. (iii) if an index $k$ is moved forward, continue from index $k + w$ (to avoid overlaps); otherwise, proceed to the next index $k + 1$.

---

**Algorithm 1** Resample-Previous-Tokens (RPT) training

---

1: **Input:** dataset $\mathcal{D}$; pretrained or initialized model $\hat{f}_\theta$ with params $\theta_0$
2: **Hyperparameters:** Probabilities $s, q \in (0,1)$; window size $w \geq 2$; number of iterations $m$
3: $\theta \leftarrow \theta_0$           ▷ Initialize parameters
4: **for** iteration $i = 1$ **to** $m$ **do**
5:      Draw $x \sim \mathcal{D}$           ▷ Draw a training sample
6:      Set $\sigma = (1, 2, \ldots, n-1)$           ▷ The identity permutation
7:      Set $\tau = (2, 3, \ldots, n)$           ▷ Next token index
8:      With probability $s$ permute $\sigma$ using $q$ and $w$           ▷ See "training" in section 2.1
9:      Compute $\tau$           ▷ Use equation 10
10:      $X = (x_\sigma, \sigma, \tau)$           ▷ Set the input to the network
11:      $Y = x_\tau$           ▷ Set the target
12:      $\mathcal{L} \leftarrow \mathcal{L}_{\text{CE}}(\hat{f}_\theta(X), Y)$           ▷ Evaluate cross-entropy loss, equation 13
13:      $\theta \leftarrow \text{optimize}(\mathcal{L})$           ▷ Update $\theta$ with optimization step
14: **end for**

---

Modeling the NTP together with the PTP conditionals (equation 12) requires the model to be aware whether the input and predicted tokens were permuted, as it cannot infer the true order of the sequence otherwise. To address this, we use a learned positional embedding layer that takes in the input (i.e., $\sigma$) and target (i.e., $\tau$) positions, e.g., for NTP, the value is always one. The model is provided with the input token $x_{\sigma_i}$, its input position $\sigma_i$, its target position $\tau_i$ (note that this is not a permutation in our case), and the output is compared to the target token using a standard cross-entropy loss:

$$\mathcal{L}(\theta) = -\mathbb{E}_{x \sim \mathcal{D}} \sum_{i=1}^{n} \log \hat{p}_\theta(x_{\tau_i} | x_{\sigma_{<i+1}}), \tag{13}$$

where $\mathcal{D}$ is the training dataset. Algorithm 1 summarizes this training procedure, which can be used to train from scratch or finetune a pretrained model.

## 3   Analysis of RPT versus NTP sampling

In this section we develop the relevant theory to compare the error in resample-previous-tokens (RPT) sampling and the standard next-token-prediction (NTP) sampling. We provide the analysis for the simple 2-token case ($w = 2$). For notational conciseness we denote by $\pi(x_i, x_{i+1}) = p(x_i, x_{i+1} | x_{<i})$ the ground-truth joint of the next two tokens $(x_i, x_{i+1})$ given the context $x_{<i}$. When there is a notational ambiguity we use a more explicit notation for joints, conditionals, and marginal, i.e., $\pi_{i,j}(x_i, x_j) = \pi(x_i, x_j)$, $\pi_{i|j}(x_i | x_j) = \pi(x_i | x_j)$, and $\pi_i(x_i) = \pi(x_i)$. We will perform an *asymptotic error* analysis. That is, we assume we introduce asymptotically small errors to the approximated/learned conditionals used in the NTP (see equation 3) and RPT (see equation 4) sampling schemes,

$$\hat{\pi}(x_i) = \pi(x_i) + \epsilon(x_i), \tag{14a}$$

$$\hat{\pi}(x_{i+1} | x_i) = \pi(x_{i+1} | x_i) + \epsilon(x_{i+1} | x_i), \tag{14b}$$

$$\hat{\pi}(x_i | x_{i+1}) = \pi(x_i | x_{i+1}) + \epsilon(x_i | x_{i+1}), \tag{14c}$$

where $\epsilon$ denotes the approximation errors and assumed to be very small (meaning that we ignore second and higher powers of $\epsilon$). The marginal $\pi_i$ and conditional $\pi_{i+1|i}$ represent the next-token-prediction (also abbreviated NTP) while $\pi_{i|i+1}$ represents the previous-token-prediction (PTP).

The two sampling methods NTP and RPT use the approximated conditionals in equation 14 and consequently introduce a certain error to their sampled joint $\epsilon_{i,i+1}$,

$$\hat{\pi}(x_i, x_{i+1}) = \pi(x_i, x_{i+1}) + \epsilon(x_i, x_{i+1}). \tag{15}$$

**Our goal**: Quantify the errors $\epsilon(x_i, x_{i+1})$ for each of the sampling method: NTP and RPT.

To quantify the *size* of the errors we will use the *total variation distance*, defined between two probability distributions $p, \hat{p}$ over a state space $\mathcal{A} = \{a\}$ by

$$\|p - \hat{p}\|_{\text{TV}} = \sup_{A \subset \mathcal{A}} |p(A) - \hat{p}(A)| = \frac{1}{2} \|p - \hat{p}\|_1, \tag{16}$$

where we denote the 1-norm of a vector $v \in \mathbb{R}^{\mathcal{A}}$ by

$$\|v\|_1 = \sum_{a \in \mathcal{A}} |v(a)|. \tag{17}$$

The RPT sampling, under certain conditions, introduces improved error compared to NTP. In more details, we define the *RPT factor* to be the *ratio* of RPT and NTP error bounds. It takes the form

**RPT factor** $\qquad \rho = \kappa \dfrac{\|\epsilon_{i|i+1}\|_\infty + \|\epsilon_{i+1|i}\|_\infty}{\|\epsilon_i\|_1 + \|\epsilon_{i+1|i}\|_\infty},$ $\qquad\qquad$ (18)

where for matrices $m \in \mathbb{R}^{\mathcal{B} \times \mathcal{A}}$ we use the max-norm

$$\|m\|_\infty = \max_a \sum_b |m(b|a)|. \tag{19}$$

**Theorem 1.** *If $\rho < 1$ then resample-previous-token (RPT) sampling achieves a lower error bound compared to a tight error bound of next-token-rediction (NPT) sampling.*

The RPT factor is composed of two parts: the first is $\kappa$ which is a function of the ground truth conditionals $\pi_{i|i+1}$ and $\pi_{i+1|i}$, which we cannot control (explained more later), and the ratio of the sum of PTP and NTP errors $\|\epsilon_{i|i+1}\|_\infty + \|\epsilon_{i+1|i}\|_\infty$ and the NTP errors $\|\epsilon_i\|_1 + \|\epsilon_{i+1|i}\|_\infty$. Our main observation is that, in practice, the PTP error $\epsilon_{i|i+1}$ is considerably lower compared to the NTP errors, $\epsilon_i$ and $\epsilon_{i+1|i}$. Intuitively, this means that, given a context $x_{<i}$, predicting the next token $x_i$ given a future token $x_{i+1}$ is easier than without it. That is

$$\underbrace{\|\epsilon_{i|i+1}\|_\infty}_{\epsilon_{\text{p}} \text{ PTP error}} < \underbrace{\|\epsilon_i\|_1}_{\epsilon_{\text{n}} \text{ NTP error}} \approx \underbrace{\|\epsilon_{i+1|i}\|_\infty}_{\epsilon_{\text{n}} \text{ NTP error}} \tag{20}$$

and consequently the second part (and the part controlled by the approximation method) of the factor is in practice smaller than 1. To justify 20 empirically we use the Pinsker inequality to bound the total variation distance of general probability distributions $p, \hat{p}$ with their KL-divergence,

$$\|p - \hat{p}\|_{\text{TV}}^2 \leq \frac{1}{2} D_{\text{KL}}(p\|\hat{p}), \tag{21}$$

where $D_{\text{KL}}(p\|\hat{p}) = H(p, \hat{p}) - H(p)$, where $H(p, \hat{p})$ is the cross entropy and $H(p)$ is the entropy. Now, making the further assumption that the entropy of the ground truth conditionals $\pi_i$, $\pi_{i|i+1}, \pi_{i+1|i}$ is near zero for non trivial context $x_{<i}$ and large vocabulary sizes $V$, we focus on the cross entropies and compare those of the NTP, i.e., $H(\pi_i, \hat{\pi}_i)$ and $H(\pi_{i+1|i}, \hat{\pi}_{i+1|i})$, and the PTP, $H(\pi_{i|i+1}, \hat{\pi}_{i|i+1})$. Figure 3 shows the cross entropies $H(\pi_i, \hat{\pi}_i)$ and $H(\pi_{i|i+1}, \hat{\pi}_{i|i+1})$ as computed during the training of a 1.5B parameter model on a validations set (training on less than 1 epoch). As can be noticed, the PTP enjoys a considerably lower cross entropy loss and therefore

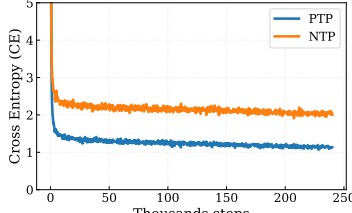

Figure 3: Cross entropy training curves of NTP and PTP.

under our assumptions also lower total variation norm. Next, we derive asymptotic bounds on the sampling errors $\epsilon$ for both NTP and RPT, and compute the RPT factor. We start with the NTP errors.

**Next-token-prediction asymptotic error** The error $\epsilon(x_i, x_{i+1})$ in the NTP sampling of the next two tokens can be calculated directly from the probability chain rule

$$\hat{\pi}(x_i, x_{i+1}) = \hat{\pi}(x_i)\hat{\pi}(x_{i+1}|x_i) \tag{22}$$

$$= \big(\pi(x_i) + \epsilon(x_i)\big)\big(\pi(x_{i+1}|x_i) + \epsilon(x_{i+1}|x_i)\big) \tag{23}$$

$$= \pi(x_i, x_{i+1}) + \epsilon(x_i)\pi(x_{i+1}|x_i) + \epsilon(x_{i+1}|x_i)\pi(x_i) + o(\epsilon), \tag{24}$$

where by $o(\epsilon)$ we denote a terms that is asymptotically smaller than $\epsilon$, e.g., in this case $\epsilon(x_i)\epsilon(x_{i+1}|x_i)$. Therefore the asymptotic error in this case is

$$\epsilon(x_i, x_{i+1}) = \epsilon(x_i)\pi(x_{i+1}|x_i) + \epsilon(x_{i+1}|x_i)\pi(x_i) + o(\epsilon). \tag{25}$$

We can use this asymptotic error expansion to derive a simple tight bound on the NTP error (see Appendix A.1 for more details),

$$\|\epsilon\|_1 \leq \|\epsilon_i\|_1 + \|\epsilon_{i+1|i}\|_\infty + o(\epsilon). \tag{26}$$

**Resample-previous-token asymptotic error** Analyzing the error $\epsilon(x_i, x_{i+1})$ in RPT sampling requires a more elaborate computation as it directly involves the *stationary distribution* of the RPT iterative sampling procedure in equation 4. The RPT iterations can be seen as a *Markov chain* where the states include all pairs of possible tokens $(x_i, x_{i+1}) \in \mathcal{V} \times \mathcal{V}$. The RPT iterations can be interpreted as sampling from the following Markov probability transition kernel, which corresponds exactly to a single iteration of equation 4,

$$p(x'_i, x'_{i+1}|x_i, x_{i+1}) = \pi(x'_i|x_{i+1})\pi(x'_{i+1}|x'_i). \tag{27}$$

Note that while $\pi(x'_{i+1}|x'_i)$ is a standard next-token-prediction, the conditional $\pi(x'_i|x_{i+1})$ predicts the $i$-th token given the context $x_{<i}$ *and* the next token $x_{i+1}$ and therefore introduces a smaller error according to the assumption in equation 20. Before estimating the asymptotic error in the joint $\epsilon(x_i, x_{i+1})$ we first need to derive the asymptotic error in the kernel. We denote this error by $e(x'_i, x'_{i+1}|x_i, x_{i+1})$ and it can be computed similar to before by expanding

$$\hat{p}(x'_i, x'_{i+1}|x_i, x_{i+1}) = \big(\pi(x'_i|x_{i+1}) + \epsilon(x'_i|x_{i+1})\big)\big(\pi(x'_{i+1}|x'_i)) + \epsilon(x'_{i+1}|x'_i)\big) \tag{28}$$

and get the kernel asymptotic error

$$e(x'_i, x'_{i+1}|x_i, x_{i+1}) = \epsilon(x'_i|x_{i+1})\pi(x'_{i+1}|x'_i) + \epsilon(x'_{i+1}|x'_i)\pi(x'_i|x_{i+1}) + o(\epsilon). \tag{29}$$

Consequently a bound on the kernel error is

$$\|e\|_\infty \leq \|\epsilon_{i|i+1}\|_\infty + \|\epsilon_{i+1|i}\|_\infty + o(\epsilon), \tag{30}$$

where we use the max-norm again; see Appendix A.2 for the exact derivation. Next, to achieve a bound on the RPT sampling error $\epsilon_{i,i+1}$ of the stationary distribution of the Markov chain 28 we can use standard perturbation bounds (Cho and Meyer, 2001; Seneta, 2006; Kirkland et al., 2008),

$$\|\epsilon\|_1 \leq \kappa \|e\|_\infty, \tag{31}$$

where $\kappa$ is a constant that depended on the ground truth Markov chain 27 known as its *conditional number*. There are different definitions of $\kappa$, some of them are tighter than others but harder to compute/provide intuitive explanation. A relatively intuitive one is $\kappa = (1 - \Lambda(P))^{-1}$. Here, $P$ is the transition matrix of the Markov process 27 with entries $p(x'_i, x'_{i+1}|x_i, x_{i+1})$, and $\Lambda(P)$ is its *ergodicity coefficient*, which is a scalar in $[0, 1]$ that quantifies how quickly the influence of the initial state diminishes over time, or equivalently, how quickly the chain converges to its stationary distribution. Plugging our kernel error (equation 30) into equation 31 we get the RPT error bound

$$\|\epsilon\|_1 \leq \kappa \big(\|\epsilon_{i|i+1}\|_\infty + \|\epsilon_{i+1|i}\|_\infty\big) + o(\epsilon). \tag{32}$$

Comparing this bound to the one in equation 26, we get the RPT factor in equation 18.

In the next paragraph we provide a synthetic experiment that shows the benefit of the RPT over NTP in a controlled setting experiment.

**Synthetic example** We have conducted a synthetic (toy) experiment to compare the NTP and RPT sampling. To that end we considered a random ground truth joint $\pi(x_i, x_{i+1})$ with $V = 20$ vocabulary size. That is, for each entry $\pi(x_i, x_{i+1})$ we sampled i.i.d. from $\mathcal{U}(0, 1)$ (uniform distribution) and normalized. To add noise to, e.g., $\pi_i$, we random a noisy distribution $n(x_i)$ (as done before) and set

$$\hat{\pi}_i(x_i) = (1 - \epsilon_n)\pi_i(x_i) + \epsilon_n n(x_i) \tag{33}$$

where $\epsilon_n > 0$ is the noise level, and similarly for the conditionals $\pi_{i+1|i}$ (with noise level $\epsilon_n$) and $\pi_{i+1|i}$ (with noise level $\epsilon_p$). We used three cases for the noise levels $\epsilon_n > \epsilon_p$: (i) *Oracle* where $\epsilon_p = 0$; (ii) *Medium* where $\epsilon_p = 0.5\epsilon_n$; and (iii) *Low* where $\epsilon_p = 0.75\epsilon_n$. We take $\epsilon_n = 1$ (we show other choices of this base noise in Appendix A.3. We numerically computed the sampling error $\epsilon_{i,i+1}$ of NTP and RPT, and repeated this experiment 1000 times. Figure 4 shows the histograms of the total variation distance $\|\epsilon_{i,i+1}\|_{\text{TV}}$ (bottom row) and max-norm $\|\epsilon_{i,i+1}\|_\infty$ (top row) per experiment, and their respective mean. Note that the smaller the ratio $\epsilon_p/\epsilon_n$ the larger the benefit in RPT sampling over NTP, and that RPT already exhibits non-trivial improvement for cases where $\epsilon_p/\epsilon_n = 0.75$.

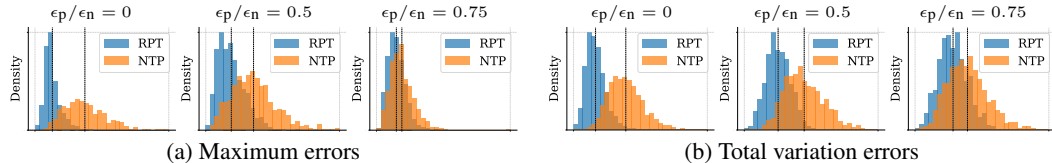

(a) Maximum errors             (b) Total variation errors

Figure 4: Synthetic sampling experiment comparing next-token-prediction (NTP) and resample-previous-token (RPT). Maximum errors and total variation errors are shown for three different error levels $\epsilon_p < \epsilon_n$. The lower the relative error $\epsilon_p/\epsilon_n$ the better RPT sampling over NTP, where RPT already shows benefit over NTP for $\epsilon_p/\epsilon_n = 0.75$.

Table 1: Results of RPT sampling on coding and reasoning task. We fine-tuned the model with an additional 100 billion tokens. Metrics are reported for both the fully autoregressively trained model (AR-F, 1 trillion tokens) and the initial checkpoint (AR-C, 900 billion tokens). Note, $k = 0$ is NTP sampling from our model. For all methods, we report best temperature in $\{0.0, 0.05, 0.1\}$.

| $k$ | HumanEval+ | MBPP | GSM8K | C++ | C# | PHP | Bash | Java | TypeScript |
|---|---|---|---|---|---|---|---|---|---|
| AR-F | 25.6 | 39.0 | 35.2 | 28.5 | 17.7 | **28.5** | 6.9 | 37.9 | 35.8 |
| AR-C | 24.4 -1.2% | 38.8 -0.2% | 35.4 +0.2% | 28.6 +0.1% | 22.8 +5.1% | 24.2 -4.3% | 8.8 +1.9% | 32.9 -5.0% | 33.9 -1.9% |
| 0 | 27.4 +1.8% | 39.6 +0.4% | 35.5 +0.3% | **31.0** +2.5% | **22.8** +5.1% | 26.1 -1.4% | **9.5** +2.6% | 38.3 +0.9% | 37.7 +1.9% |
| 0.5 | 27.4 +1.8% | 39.6 +0.4% | 37.3 +2.1% | **31.0** +2.5% | **22.8** +5.1% | 26.1 -1.4% | 9.5 +2.6% | 40.5 +2.6% | 37.7 +1.9% |
| 1 | 27.4 +1.8% | **40.6** +1.4% | **37.5** +2.3% | **31.0** +2.5% | 22.2 +4.5% | 25.5 -3.0% | 9.5 +2.6% | **41.1** +3.2% | **38.4** +2.6% |
| 1.5 | **28.6** +3.0% | **40.6** +1.4% | **37.5** +2.3% | **31.0** +2.5% | 22.2 +4.5% | 25.5 -3.0% | 9.5 +2.6% | 41.1 +3.2% | **38.4** +2.6% |
| 2 | 28.0 +2.4% | 39.0 +0.0% | 36.3 +1.1% | **31.0** +2.5% | 19.6 +1.9% | 26.7 -1.8% | 9.5 +2.6% | 37.9 +0.0% | 37.7 +1.9% |

## 4 Related work

**Sampling methods** Autoregressive models typically employ NTP sampling, which is based on the chain rule of probabilities. However, NTP sampling has a limitation: it cannot correct previously predicted tokens, potentially propagating errors throughout the generated sequence. To address this issue, prior works have explored searching the model's probability space using techniques like Beam Search (Freitag and Al-Onaizan, 2017). Despite their appeal, these methods are not widely adopted in practice due to issues like repetition and collapse (Holtzman et al., 2020). In contrast, our work extends the traditional NTP sampling by introducing RPT, a novel sampling approach that allows arguably improved sampling from the joint probability of tokens.

**Predictor-corrector** Discrete diffusion models often apply remasking of previously revealed tokens (Lezama et al., 2022; Campbell et al., 2024; Gat et al., 2024), but these methods are not be directly applicable to autoregressive LLMs. Alternatively, other works suggest prompting the model to self-correct through carefully designed inputs (Chen et al., 2025; Muennighoff et al., 2025). Other approaches modify the model architecture or generation process to achieve similar goals: Li et al. (2024) introduce a predictor-corrector mechanism by modifying the transformer architecture to accumulate states; Stern et al. (2019); Gu et al. (2019) enable insertion, deletion, and replacement of tokens during generation, effectively performing implicit corrector iterations. While previous works have made significant progress, they fall short of achieving state-of-the-art performance. In this work, we propose novel training and sampling procedures that enable corrector iterations for LLMs, improving the current state-of-the-art.

**Any-order architectures** Our method augments standard next-token-prediction with explicit information about permutations of the input and target tokens. Prior work has explored any-order language modeling. Pannatier et al. (2024) propose an architecture that uses double positional embeddings to encode the absolute positions of both the observed tokens and the targets to be predicted. Yang et al. (2019) achieve permutation awareness by permuting the attention bias. A complementary line of research employs discrete flow or diffusion models, which—by virtue of their bidirectional attention and path training objectives can naturally generate tokens in arbitrary orders (Lezama et al., 2022; Gat et al., 2024; Lipman et al., 2024; Shaul et al., 2024; Holderrieth et al., 2025; Nie et al., 2025). In contrast to these global-order methods, our sampler requires only local permutation awareness within a window of size $w$, and uses non permutation targets $\tau$. This locality requires only a small embedding table of dimension $w + 1$ that encodes the *relative* order of the $w$ most recent context tokens. This simple addition preserves the conventional next-token-prediction loss while requiring only a minor modification to the training pipeline.

# 5 Experiments

We study the performance of our resample-previous-token (RPT) in practice. First, in Section 5.2 we demonstrate that RPT sampling can be trained with a minimal fine-tuning of a pretrained next-token-prediction (NTP) model, while maintaining its NTP performance. Second, in Section 5.3 we perform an error analysis, comparing RPT to traditional NTP sampling showing improvement in total variation distance of learned conditional probabilities $\hat{p}(x_i|x_{<i})$ measured over test data, as partially predicted by the theory in Section 3. Finally, in Section 5.4 we experiment with RPT sampling on code generation and reasoning benchmarks demonstrating improvements in both RPT pretraining and sampling.

## 5.1 Implementation details

**Data, architectures, training, and baselines** Our data consists of a corpus of one trillion (1T) language tokens. Throughout all experiments, we use the same training dataset and maintain the same data order. We pretrained an autoregressive model with 8B parameters, using the standard cross entropy loss (i.e., equation 13 with $\sigma_i = i$ and $\tau_i = i + 1$) and the same architectural design as in Meta (2024) on this 1T token data for 240K iterations as our baseline, denoted **AR-F**. We denote its 224K iteration checkpoint (i.e., after 90% of the training tokens) by **AR-C**. We next finetuned AR-C to reach 240K iteration and the remaining 100B tokens (10% of total training tokens) with $m = 16K$ iterations in Algorithm 1 with window size $w = 3$, and hyper-parameters $s = 0.5$ and $q = 0.02$, which corresponds to 80 expected swaps in each sequences of $n = 4096$ tokens (equation 9). We train on 256 H100 GPUs and a batch size of 4M tokens. We use AdamW optimizer with a warmup of 2000 steps, a peak learning rate of 1e-3 and a cosine scheduler.

**Positional encoding** Our method requires information on whether the predicted token is permuted, see line 10 and 12 in Algorithm 1. To incorporate this information, we introduce a learned positional embedding layer. One option is to learn this layer applied to the source-target pair, $(\sigma_i, \tau_i)$, however this incorporates global positional encoding. Inspired by relative positional encoding (Su et al., 2021) we note that the source-target pair can be encoded relatively by using the *difference* $\tau_i - \sigma_i$ as input to the learned positional embedding layer. For example, in Figure 2, the positional embeddings are $\{1, 1, -1, 2, 1, \dots, 1\}$.

**Practical sampling** We focus on sampling with $w = 2$ in the paper as we didn't see a practical benefit in sampling with $w = 3$ window size, which is more computationally demanding (see Appendix B.1). A benefit in RPT sampling algorithms is that it allows incorporating different sampling heuristics not usually available in standard AR sampling. Two useful heuristic we tested are: (i) *Greedy decoding*: using $\arg\max$ in PTP sampling $\hat{p}(x_i|x_{i+1}, x_{<i})$; and (ii) *Confidence*: accepting a token only if its probability is greater than a threshold, i.e., $\eta = 0.9$. In the main paper we always use Greedy decoding; we show ablation on Greedy decoding in Appendix B.2.

## 5.2 Evaluation

We first verify that our fine-tuning process does not degrade the NTP sampling performance. Figure 5 presents the autoregressive training loss for 900B tokens and the fine-tuning phase, where we train our proposed method for an additional 100B tokens. The NTP error continues the same trend where, at convergence, we observe a slightly higher NTP loss ($\sim$0.02), which is negligible in terms of cross-entropy (compared to 1.75 at convergence). At the same time the PTP losses (there are two such conditionals in $w = 3$ training corresponding to $\ell = 1, 2$ in equation 8) rapidly converge to a substantially lower error values compared to the NTP loss, as conjectured in our theoretical analysis in equation 20 in Section 3. Intuitively, the more future tokens are used for next-token-prediction, the better. Moreover, in Table 1, we show that sampling with $k = 0$ (NTP sampling with the fine-tuned model), often achieves better scores on the benchmarks compared to our autore-

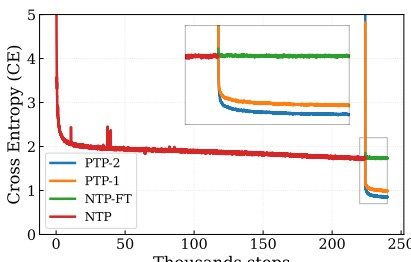

Figure 5: NTP losses of the pretrain (in red) and fine-tune (in green) stages. We also show the loss curves of PTP-1 (in orange) and PTP-2 (in blue) that correspond to $\ell = 1$ and $\ell = 2$ in eq. (8).

gressive bassline at convergence, AR-F. This suggests that our fine-tuning in fact could potentially improve the model's performance.

## 5.3 Error analysis

We compute the empirical total variation (TV) distance (equation 16) between the conditional $\hat{p}(x_i|x_{<i})$ as computed by RPT, and the ground truth distribution $p(x_i|x_{<i})$ represented by the ground truth token $x_i$ in a validation set. In Table 2 we report empirical TV distances computed with 128K validation set tokens from each dataset, all of them with context of at-least 20 tokens, and compute the RPT error $\epsilon^{(k)}(x_i) = 1 - \hat{p}^{(k)}(x_i|x_{<i})$ where $\hat{p}^{(k)}(x_i|x_{<i})$ is the predicted probability of $x_i$ after $k = 0, 1, 2, 3$ RPT iterations (see 4). Iteration $k = 0$ corresponds to NTP

Table 2: Empirical total variation distance (lower is better) of RPT conditionals measured on validation data for an increasing number of RPT iterations $k$.

| $k$ | dclm | github | wiki | arxiv | se |
|---|---|---|---|---|---|
| 0 | .34 | .18 | .26 | .27 | .27 |
| 1 | .30 | .15 | .23 | .23 | .24 |
| 2 | .30 | .14 | .23 | .22 | .23 |
| 3 | .30 | .14 | .22 | .22 | .23 |

sampling with our trained model, and the TV error decreases mostly after the first RPT iteration, and continue to decrease moderately afterwards; this is consistent across all tested validation datasets. The analysis in Section 3 discusses the error in the limit (mixed) case, which seems to happen after only a few iterations in practice (for our chosen sampling method).

Figure 6 compares the probabilities assigned to ground truth tokens $x_i$ by RPT (after $k = 1$ iteration) and NTP sampling process, i.e., $\hat{p}^{(1)}(x_i) - \hat{p}^{(0)}(x_i)$ on the DCLM dataset. RPT improves the probability of the ground truth token in 64.5% cases which is another indication of the improved sampling properties fo RPT compared to NTP.

Figure 6: Difference of RPT ($k = 1$) and NPT probabilities on validation tokens. RPT improves probabilities of validation tokens.

## 5.4 Coding and reasoning tasks

We evaluate our method on popular coding and reasoning benchmarks: HumanEval+ (Chen et al., 2021; Liu et al., 2023) is a task where the model is required to complete a given a function signature with a docstring. MBPP (Austin et al., 2021) contains few-shot code generation tasks from problem descriptions. GSM8K (Cobbe et al., 2021) consists of grade-school-level mathematical word problems. Finally, we report results on MultiPL-E, a non-Pythonic version of HumanEval (Ben Allal et al., 2022).

Table 1 summarizes the results of our experiments, comparing the performance of RPT sampling to NTP on several baselines. AR-F is our fully trained AR model using NTP sampling. We also report results using the initial fine-tuning checkpoint AR-C, providing a direct comparison to the initial pre-fine-tuning state. Additionally, we include results from NTP sampling on our trained model ($k = 0$). Our results demonstrate a consistent and significant improvement in performance when using RPT sampling compared to the NTP baseline. RPT improves both in pretraining (0 iterations) and RPT sampling ($>0$ iterations) rather consistently across all 9 benchmarks. All results are reported with confidence sampling. In Appendix B.2 we also provide the ablation results for non-confidence sampling, and non-greedy sampling. For a fair comparison between the methods, we follow Gloeckle et al. (2024) and report oracle score over the temperatures $\{0.0, 0.05, 0.1\}$.

# 6 Conclusions and future work

We introduce Resample-Previous-Tokens (RPT), a sampling method that allows models to revisit and replace previously generated tokens. After a short fine-tuning of a pretrained model, RPT leads to approximately 10% relative improvements in reasoning and coding tasks. In the paper, we analyze the method in a controlled environment as well as large-scale validation datasets. We believe that RPT sampling has the potential to further enhance AR performance by incorporating larger window sizes and other pre-defined permutations, such as block-permutations. As our work introduces an alternative sampling paradigm to autoregressive models, it does not seem to introduce significant societal risks beyond those that already exist with large language models.

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

# A Analysis of RPT versus NTP sampling

## A.1 Next-token-prediction asymptotic error

Let us first derive the NTP error bound

$$\|\epsilon\|_1 \le \|\epsilon_i\|_1 + \|\epsilon_{i+1|i}\|_\infty + o(\epsilon). \tag{34}$$

consider equation 25, i.e.,

$$\epsilon(x_i, x_{i+1}) = \epsilon(x_i)\pi(x_{i+1}|x_i) + \epsilon(x_{i+1}|x_i)\pi(x_i) + o(\epsilon). \tag{35}$$

and sum both sides after taking absolute values and using the triangle inequality on the r.h.s. to get

$$\|\epsilon_{i,i+1}\|_1 \le \sum_{x_i, x_{i+1}} |\epsilon(x_i)|\pi(x_{i+1}|x_i) + |\epsilon(x_{i+1}|x_i)|\pi(x_i) + o(\epsilon^2) \tag{36}$$

$$= \sum_{x_i} |\epsilon(x_i)| + \sum_{x_i} \pi(x_i) \left( \sum_{x_{i+1}} |\epsilon(x_{i+1}|x_i)| \right) \tag{37}$$

$$\le \|\epsilon_i\|_1 + \|\epsilon_{i+1|i}\|_\infty. \tag{38}$$

This bound will be tight for any scenario where

$$\epsilon(x_i)\epsilon(x_{i+1}|x_i)\pi(x_i)\pi(x_{i+1}|x_i) \ge 0. \tag{39}$$

We provide a simple example when this holds. Consider $V = 2$ (vocabulary of size 2), and let

$$\pi = \begin{bmatrix} \alpha & 0 \\ 0 & \beta \end{bmatrix}, \qquad \epsilon = \begin{bmatrix} \epsilon & 0 \\ 0 & -\epsilon \end{bmatrix}, \qquad \hat{\pi} = \begin{bmatrix} \alpha + \epsilon & 0 \\ 0 & \beta - \epsilon \end{bmatrix}, \tag{40}$$

with $\alpha, \beta > 0$ and $\alpha + \beta = 1$ and $0 < \epsilon < \min\{a, b\}$. In this case we have $\pi(x_{i+1}|x_i) = \hat{\pi}(x_{i+1}|x_i)$ for all $x_i, x_{i+1}$ and therefore $\epsilon(x_{i+1}|x_i) = 0$ and equation 39 holds trivially.

## A.2 Resample-previous-token asymptotic error

We derive the asymptotic error in the RPT Markov chain transition kernel and consequently the error bound in equation 32. First, the asymptotic kernel error is

$$\hat{p}(x_i', x_{i+1}'|x_i, x_{i+1}) = \hat{\pi}(x_i'|x_{i+1})\hat{\pi}(x_{i+1}'|x_i')$$
$$= \left(\pi(x_i'|x_{i+1}) + \epsilon(x_i'|x_{i+1})\right)\left(\pi(x_{i+1}'|x_i') + \epsilon(x_{i+1}'|x_i')\right)$$
$$= p(x_i', x_{i+1}'|x_i, x_{i+1}) + \epsilon(x_i'|x_{i+1})\pi(x_{i+1}'|x_i') + \epsilon(x_{i+1}'|x_i')\pi(x_i'|x_{i+1}) + o(\epsilon),$$

where $o(\epsilon)$ are asymptotically smaller than $\epsilon$ terms. Therefore the asymptotic kernel error is

$$e(x_i', x_{i+1}'|x_i, x_{i+1}) = \epsilon(x_i'|x_{i+1})\pi(x_{i+1}'|x_i') + \epsilon(x_{i+1}'|x_i')\pi(x_i'|x_{i+1}) + o(\epsilon). \tag{41}$$

Let us compute the max-norm of the kernel errors,

$$\|e\|_\infty = \max_{x_i, x_{i+1}} \sum_{x_i', x_{i+1}'} |e(x_i', x_{i+1}'|x_i, x_{i+1})| \tag{42}$$

$$\le \max_{x_i, x_{i+1}} \left\{ \sum_{x_i', x_{i+1}'} |\epsilon(x_i'|x_{i+1})|\, \pi(x_{i+1}'|x_i') + \sum_{x_i', x_{i+1}'} |\epsilon(x_{i+1}'|x_i')|\, \pi(x_i'|x_{i+1}) \right\} + o(\epsilon) \tag{43}$$

$$\le \max_{x_i, x_{i+1}} \left\{ \sum_{x_i'} |\epsilon(x_i'|x_{i+1})| + \sum_{x_i'} \left( \sum_{x_{i+1}'} |\epsilon(x_{i+1}'|x_i')| \right) \pi(x_i'|x_{i+1}) \right\} + o(\epsilon) \tag{44}$$

$$\le \max_{x_{i+1}} \sum_{x_i'} |\epsilon(x_i'|x_{i+1})| + \max_{x_i'} \sum_{x_{i+1}'} |\epsilon(x_{i+1}'|x_i')| + o(\epsilon) \tag{45}$$

$$= \left\|\epsilon_{i|i+1}\right\|_\infty + \left\|\epsilon_{i+1|i}\right\|_\infty + o(\epsilon). \tag{46}$$

## A.3 Synthetic example - base noises

In section 3 we present results with $\epsilon_n = 1$. Here we report results of our synthetic experiment with different base noises, $\epsilon_n \in \{0.1, 0.01\}$. The RPT improvements are invariant to absolute noise level.

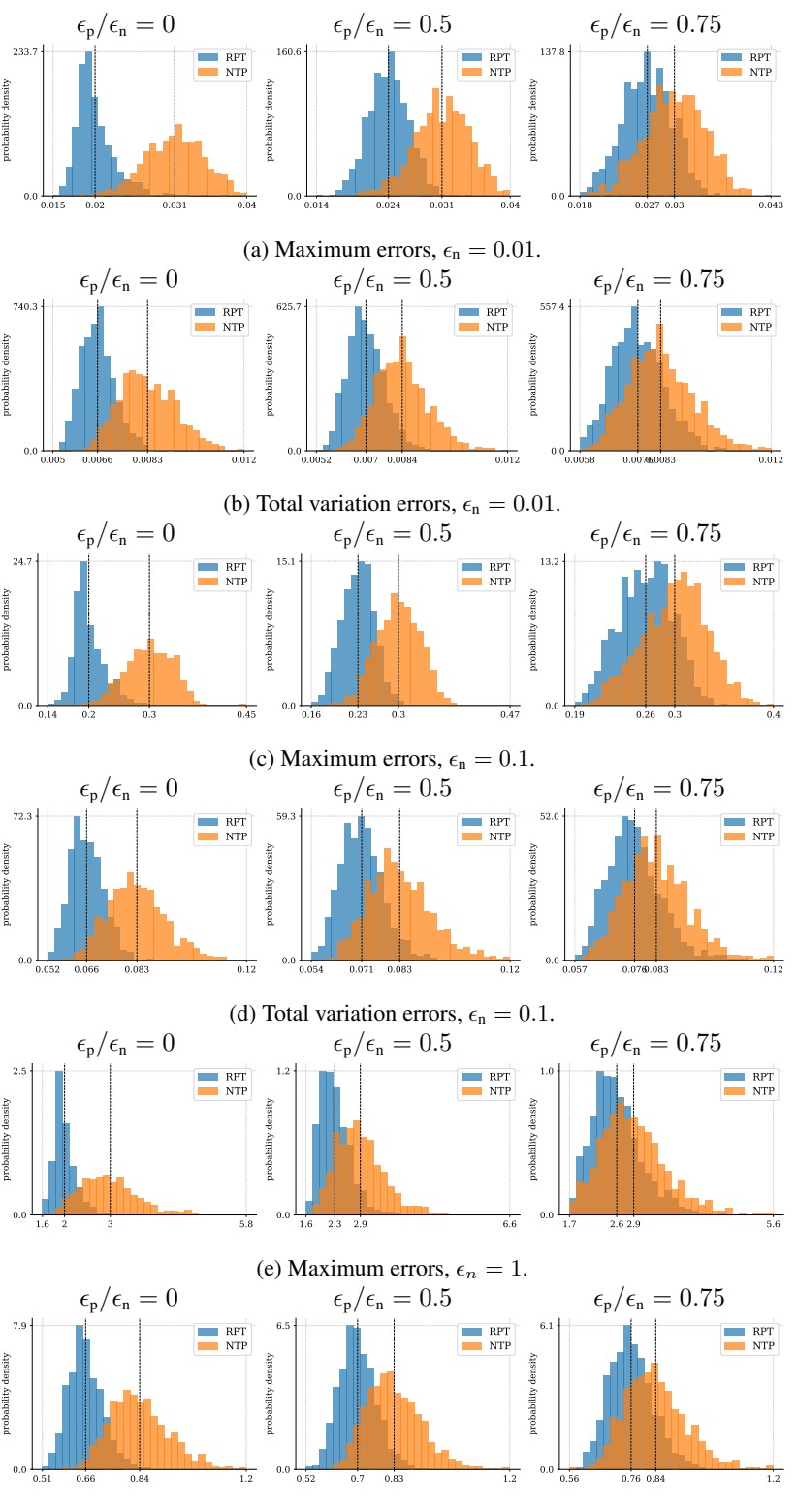

(a) Maximum errors, $\epsilon_n = 0.01$.

(b) Total variation errors, $\epsilon_n = 0.01$.

(c) Maximum errors, $\epsilon_n = 0.1$.

(d) Total variation errors, $\epsilon_n = 0.1$.

(e) Maximum errors, $\epsilon_n = 1$.

(f) Total variation errors, $\epsilon_n = 1$.

# B  Ablations

## B.1  Window size

Our trained model supports a sampling window of three tokens ($w = 3$). In the main paper, we present results for a two-token window ($w = 2$) sampling. A comparison between the two window sizes is presented in the table below, which shows that sampling with two and three tokens yields roughly comparable results. Given the similar performance and the increased efficiency of the two-token window during sampling, we opted to present results using the two-token window in the main paper.

| $k$ | $w$ | HumanEval+ | MBPP | GSM8K | C++ | C# | PHP | Bash | Java | TypeScript |
|---|---|---|---|---|---|---|---|---|---|---|
| AR-F | - | 25.6 | 39.0 | 35.2 | 28.5 | 17.7 | 28.5 | 6.9 | 37.9 | 35.8 |
| AR-C | - | 24.4 | 38.8 | 35.4 | 28.6 | 22.8 | 24.2 | 8.8 | 32.9 | 33.9 |
| 0 | - | 27.4 | 39.6 | 35.5 | **31.0** | **22.8** | 26.1 | **9.5** | 38.3 | 37.7 |
| 1 | 2 | 27.4 | **40.6** | **37.5** | **31.0** | 22.2 | 25.5 | **9.5** | 41.1 | **38.4** |
| 2 | 2 | 28.0 | 39.0 | 36.3 | **31.0** | 19.6 | **26.7** | **9.5** | 37.9 | 37.7 |
| 1 | 3 | 28.0 | 39.2 | 35.5 | 30.4 | 22.2 | 25.5 | **9.5** | **41.7** | **38.4** |
| 2 | 3 | **28.6** | 39.2 | 35.6 | 30.4 | 19.6 | **26.7** | **9.5** | 39.2 | **38.4** |

## B.2  Sampling parameters

Table 1 reports results on various task benchmarks using our default RPT sampling that includes greedy decoding and confidence (set to 0.9), see the practical sampling paragraph in Section 5.1. Here we complete the picture by showing in the table below ablation results of samplings without greedy sampling and confidence. We find that the chosen setup is the most stable among the options considered. Note however, that naive sampling already introduces non-trivial improvements.

| $k$ | HumanEval+ | MBPP | GSM8K | C++ | C# | PHP | Bash | Java | TypeScript |
|---|---|---|---|---|---|---|---|---|---|
| AR-F | 25.6 | 39.0 | 35.2 | 28.5 | 17.7 | 28.5 | 6.9 | 37.9 | 35.8 |
| AR-C | 24.4 | 38.8 | 35.4 | 28.6 | 22.8 | 24.2 | 8.8 | 32.9 | 33.9 |
| 0 | 27.4 | 39.6 | 35.5 | 31.0 | **22.8** | 26.1 | **9.5** | 38.3 | 37.7 |
| 1 | 28.6 | 39.0 | 36.0 | **32.3** | 20.9 | **27.9** | 7.0 | 34.8 | 37.7 |
| 2 | **29.9** | 39.0 | 35.5 | 30.4 | 22.2 | **27.9** | 8.9 | **41.1** | 35.2 |
| + Greedy decoding | | | | | | | | | |
| 1 | **29.9** | 38.8 | 35.0 | 31.7 | 19.0 | **27.9** | 7.0 | 37.6 | 35.2 |
| 2 | 28.6 | 38.8 | 35.6 | **32.3** | 20.9 | **27.9** | 7.0 | 37.6 | 35.2 |
| + Greedy decoding + confidence | | | | | | | | | |
| 1 | 27.4 | **40.6** | **37.5** | 31.0 | 22.2 | 25.5 | **9.5** | **41.1** | **38.4** |
| 2 | 28.0 | 39.0 | 36.3 | 31.0 | 19.6 | 26.7 | **9.5** | 37.9 | 37.7 |

# C Temperature analysis

We compare RPT sampling ($k > 0$) to NTP sampling ($k = 0$) using nucleus sampling (Holtzman et al., 2020) with a top-p value of 0.95 and various sampling temperatures. The figures below suggest that our method is highly and consistently effective, particularly at reasonably large temperatures, which are larger than the commonly used temperature values for pass@1 evaluations. We demonstrate these results across various sampling parameter choices (see Appendix B) and find that RPT sampling consistently improves performance compared to NTP sampling. Experiments here are performed on HumanEval+.

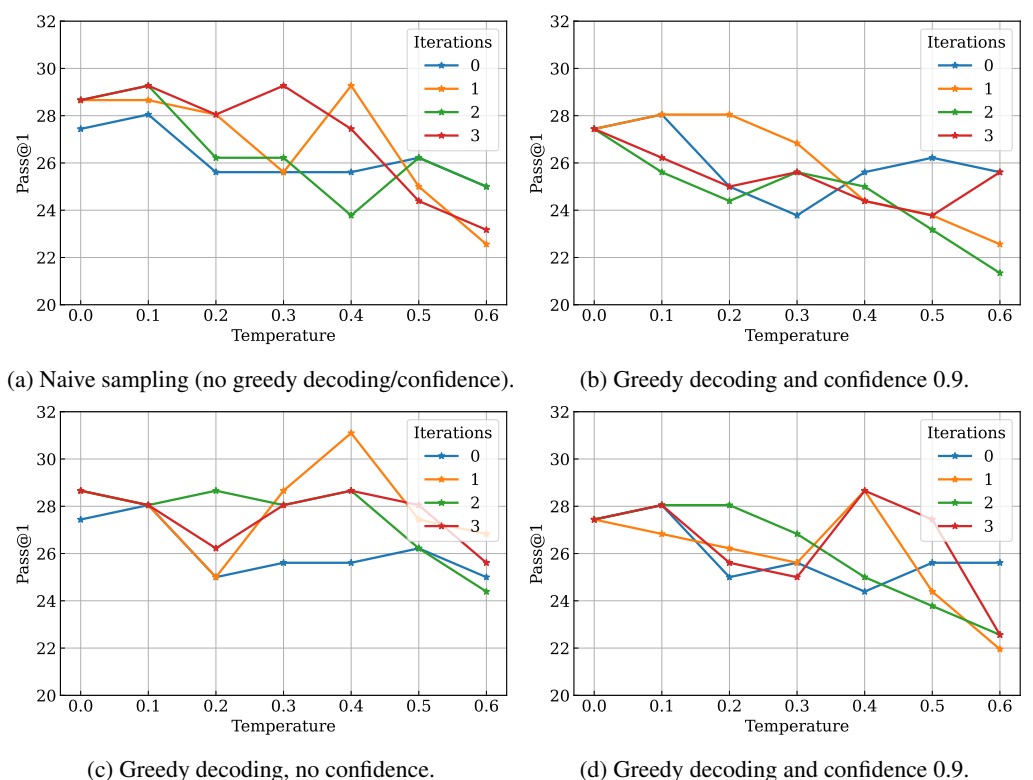

(a) Naive sampling (no greedy decoding/confidence).

(b) Greedy decoding and confidence 0.9.

(c) Greedy decoding, no confidence.

(d) Greedy decoding and confidence 0.9.

