# OpenReview forum: "Corrector Sampling in Language Models"
_NeurIPS.cc/2025/Conference — NeurIPS 2025 poster_

### Official Review · Reviewer_eT7X · 2025-06-15

**Clarity:** 3
**Significance:** 4
**Originality:** 3
**Rating:** 4
**Confidence:** 4

**Summary:**

This paper introduces Resample-Previous-Tokens (RPT), a new sampling method for autoregressive language models. RPT iteratively revisits and replaces tokens in a window to reduce error accumulation. Fine-tuning an 8B model with RPT for 100B tokens improves reasoning and coding benchmarks by 10%. RPT preserves next-token-prediction quality and speed, outperforming standard sampling.

**Questions:**

1. What does the parameter n in Table 1 represent?
2. Llama-3 uses RoPE, which already has relative position information. Why still need to add a new positional encoding? Can you directly set the position to $\sigma$?
3. Why do the authors claim "preserving their next-token-prediction speed"? During inference, each token needs to be predicted at least twice. Does this mean that the decoding speed is twice as fast?
4. In the Appendix. B, $\omega=3$ is used for training, but $\omega=2$ works better for inference. Is this in line with expectations? Does this mean that the scaling limit of this method is very low? After all, in code and mathematical reasoning scenarios, a large number of tokens often need to be modified.

**Ethical Concerns:**

["NO or VERY MINOR ethics concerns only"]

**Final Justification:**

Considering that the effect of this method when there are many tokens to be modified has not been verified, I will keep my score unchanged.

**Limitations:**

see weaknesses

**Quality:**

3

**Strengths And Weaknesses:**

#### Strengths
1. The paper proposes a good solution to save the token consumption of large models
2. RPT maintains next-token prediction quality.
3. Both theory and experiment have proved that the upper limit of RPT is higher than that of NTP.

#### Weaknesses
1. The processing of position coding is relatively rough, the sequence order is modified, but no ablation experiment is performed on position coding.
2. The paper does not provide a larger scaling. Can the mode modify the previous 512 tokens?

---

> ### Author Rebuttal · Authors · 2025-07-30
>
> > The processing of position coding is relatively rough, the sequence order is modified, but no ablation experiment is performed on position coding.
>
> By rough as assume the reviewer means one of two things: 1) the positional encoding is costly; or 2) training-inference discrepancy (similar to reviewer 6ySe).
>
> For 1) note that the relative positions is added only to the token embedding layer, and introduces a very small additional parameter count: 12k parameters for relative positions for $w=2$ and an 8B model, versus 500m parameters used by the token embedding layer. In terms of training complexity, only $O(w)$ new vectors are added to the embedding layers, which is negligible compared to vocabulary size. During inference, this additional lookup table computation is parallelized with the token embedding layer and does not introduce any overhead. We will add this discussion to the revised manuscript.
>
> For 2) note that the training and inference of RPT are performed  **exactly the same way** and therefore there is no training-inference discrepancy.
>
> > The paper does not provide a larger scaling. Can the model modify the previous $512$ tokens?
>
> While training with large window sizes (e.g., $w=512$) is feasible, it leads to very intensive inference iterations, which is why it was not tested.
>
> > What does the parameter n in Table 1 represent?
>
> The parameter $n$ in Table 1 is a typo and should be $k$. We will correct this in the revised manuscript.
>
> > Llama-3 uses RoPE, which already has relative position information. Why still need to add a new positional encoding? Can you directly set the position to $\sigma$?
>
> The RPT sampling requires the model to receive the extra information of which token it needs to predict next, which is not necessarily the next token. For example, for a window of size $w=2$ it can be the next token (with positional encoding “+1”), or the previous token (with positional encoding “-1”). RoPE works with input token locations rather than target locations and hence does not cover this use case.
>
>
> > Why do the authors claim "preserving their next-token-prediction speed"? During inference, each token needs to be predicted at least twice. Does this mean that the decoding speed is twice as fast?
>
> Reading back this sentence we agree it is confusing. Our claim of “preserving next-token-prediction speed” (and quality) refers specifically to the standard autoregressive generation process using our model, where no corrector-iterations are applied. We will remove this sentence from the revised version.
>
> > In the Appendix. B, $w=3$ is used for training, but $w=2$ works better for inference. Is this in line with expectations? Does this mean that the scaling limit of this method is very low? After all, in code and mathematical reasoning scenarios, a large number of tokens often need to be modified.
>
> Indeed, sampling with $w=3$ (Appendix B.1) achieved comparable results to $w=2$ (Table 1, main paper) however $w=2$ is a faster sampling method, so generally preferable. We do believe that it is quite possible that in other benchmarks $w\geq 3$ could improve over $w=2$, but for the benchmarks we tested in this paper that was not the case. An interesting future research venue could be to replace single tokens with blocks of tokens (or more generally, patterns of tokens) which can also have useful applications to code and math reasoning.

---

### Official Review · Reviewer_6ySe · 2025-07-01

**Clarity:** 3
**Significance:** 2
**Originality:** 3
**Rating:** 4
**Confidence:** 3

**Summary:**

This paper introduces a new sampling method, resample previous token, to allow the LLM to correct the previous tokens by themselves. The authors design a new training objective by modifying the positions/orders of tokens in a sequence to make it compatible with existing training infrastructure for next token prediction in autoregressive model. The authors provide theoretical analysis on why the correction prediction (conditioned on the future token) is easier than standard next token prediction. The experiments on coding benchmark show that the proposed method improves over the next-token-prediction baseline in pretraining.

**Questions:**

- What is the definition of $\hat{p}$ and $p$ in Eq.(4)? Is the former an approximation and the latter the ground-truth distribution? It will be more clear if there are some explanations.
- Most evaluation comparisons are on coding and math (only GSM8K). What's the performance of proposed method on other domains? E.g., general QA, science.
- Why do you use 90% data for normal next-token prediction training and only 10% for RPT? Do you have any ablation of data proportion on final performances? For example, what if you use the full dataset for RPT training?
- In general, how long will it take in RPT inference compared to NTP for window size=2?

**Ethical Concerns:**

["NO or VERY MINOR ethics concerns only"]

**Final Justification:**

My major concern on training-inference gap is resolved so I will raise my score to 4.

**Limitations:**

yes.

**Quality:**

2

**Strengths And Weaknesses:**

**strengths**
- Overall, the paper is well written and the illustrations such as fig.1 and 2 are clear.
- The idea of RPT sampling to allow the LLM to self-correct the generated token is novel. Moreover, the authors also explain the effectiveness by an asymptotic analysis on approximation error.

**weaknesses**
- My major concern on this paper is on the training paradigm of RPT: In inference, RPT corrects token in the middle (e.g., when predicting $x_3$ by $x_1,x_2, x_4$, the $x_1,x_2,x_4$ are at the positions $1,2,4$). However, in training, RPT uses different positions of tokens after the corrected token (e.g., the $x_1,x_2,x_4$ are at the positions $1,2,3$). Meanwhile, the training objective of $x_5$ also changes in fig.2: the input becomes $x_1,x_2,x_4,x_3$ instead of $x_1,x_2,x_3,x_4$. Although the authors introduce new position embedding, I still feel there is large gap between training and inference.
- In experiment, the authors train the model by perturbing data with window size=3. However, the table in Appendix B.1 shows that window size =3 achieves similar or even worse performances than window size =2, which also suggests the training inference gap.

---

> ### Author Rebuttal · Authors · 2025-07-30
>
> > My major concern on this paper is on the training paradigm of RPT: In inference, RPT corrects token in the middle (e.g., when predicting $x_3$ by $x_1,x_2, x_4$, the $x_1,x_2,x_4$ are at the positions $1,2,4$). However, in training, RPT uses different positions of tokens after the corrected token (e.g., the $x_1,x_2,x_4$ are at the positions $1,2,3$). Meanwhile, the training objective of $x_5$ also changes in fig.2: the input becomes $x_1,x_2,x_4,x_3$ instead of $x_1,x_2,x_3,x_4$. Although the authors introduce new position embedding, I still feel there is a large gap between training and inference.
>
> We believe there is a misunderstanding here. The training and inference of RPT are performed  **exactly the same way**. For the example in Figure 2: assuming we have $x_1,x_2,x_3,x_4$ and want to resample $x_3$ we push the **permuted sequence** $(x_1,x_2,x_4)$ (**in positions $1,2,3$, exactly as in training**, along with the positional encoding) through the network to get a prediction for $x_3$.
>
> To clarify this we will add the following explicit inference pseudo-code to the revised paper.
>
> | |
> | :--- |
> | Require: Trained parameters $\theta$, prompt $x_{<1}$, window size $w$, number of iterations $k$. |
> | Sample with NTP $x_1,\ldots,x_{w-1}$ |
> | **For** $i = w, w+1, \dots, n$ |
> | &nbsp;&nbsp;&nbsp;&nbsp;Sample with NTP $x_i$ |
> | &nbsp;&nbsp;&nbsp;&nbsp;**For** $j = 1, \dots, k$ |
> | &nbsp;&nbsp;&nbsp;&nbsp;&nbsp;&nbsp;&nbsp;&nbsp;**For** $s = w-1, w-2, \dots, 0$ &nbsp;&nbsp;&nbsp;&nbsp;% we want to replace $x_{i-s}$ |
> | &nbsp;&nbsp;&nbsp;&nbsp;&nbsp;&nbsp;&nbsp;&nbsp;&nbsp;&nbsp;&nbsp;&nbsp;Create permutation &nbsp;&nbsp;&nbsp;&nbsp;&nbsp;$\sigma = (1, 2, \dots, i) \setminus \{i-s\}$ |
> | &nbsp;&nbsp;&nbsp;&nbsp;&nbsp;&nbsp;&nbsp;&nbsp;&nbsp;&nbsp;&nbsp;&nbsp;Create target index &nbsp;&nbsp;&nbsp;&nbsp;&nbsp;&nbsp;$\tau = (2, 3, \dots, i-s)$ |
> | &nbsp;&nbsp;&nbsp;&nbsp;&nbsp;&nbsp;&nbsp;&nbsp;&nbsp;&nbsp;&nbsp;&nbsp;Create permuted sequence $x_\sigma = (x_{\sigma_1}, \dots, x_{\sigma_i})$|
> | &nbsp;&nbsp;&nbsp;&nbsp;&nbsp;&nbsp;&nbsp;&nbsp;&nbsp;&nbsp;&nbsp;&nbsp;Feed into model $(x_\sigma, \tau-\sigma)$ and sample $x_{i-s}$ |
>
> Regarding the $x_5$ target in the example of Figure 2: Indeed in each block there is one token that predicts $w$ locations into the future. One could mask out this token in the cross-entropy loss, which we indeed tested while working on the submission and found it has no influence on results. This can be explained by noting that in this case the model gets a unique relative positional encoding of $w$ for these tokens, see Section 5.1.
>
>
> > In experiment, the authors train the model by perturbing data with window size=3. However, the table in Appendix B.1 shows that window size =3 achieves similar or even worse performances than window size =2, which also suggests the training inference gap.
>
> First, as mentioned above, we respectfully claim there is **no training-inference gap**. Training with $w=3$ inherently covers all cases $w≤3$. Indeed, we found that $w=2$ and $w=3$ yield comparable performance on benchmarks. We highlight $w=2$ as it is more computationally efficient, requiring fewer forward passes during inference and therefore was overall favorable and made it into the main paper.
>
> > What is the definition of $\hat{p}$ and $p$ in Eq.(4)? Is the former an approximation and the latter the ground-truth distribution? It will be more clear if there are some explanations.
>
> Yes. $p$ is the ground truth join distribution and $\hat{p}$ is the distribution learned by the model. We define the hat operator in L40 (see second parentheses where we define $\hat{*}$) and $p$ in L41. If the reviewer feels more clarification is required, we are happy to elaborate in the revised paper.
>
> > Most evaluation comparisons are on coding and math (only GSM8K). What's the performance of the proposed method on other domains? E.g., general QA, science.
>
> To address the reviewer's comment, we evaluated our model on additional question-answering datasets (TQA and NQ). The table below presents results consistent with the coding and reasoning findings from the original submission. We will incorporate these new results into the revised paper.
>
>
> | Iterations | TQA | NQ |
> |---|---|---|
> | AR-F | 61.48 | 31.65 |
> | AR-C | 61.30 (-0.18%) | 31.53 (-0.12%) |
> | 0 | 61.64 (+0.16%) | 32.01 (+0.48%) |
> | 1 | 62.85 (+1.55%) | 33.30 (+1.77%) |
> | 2 | 63.13 (+1.83%) | 33.41 (+2.11%) |
>
> > Why do you use 90% data for normal next-token prediction training and only 10% for RPT? Do you have any ablation of data proportion on final performances? For example, what if you use the full dataset for RPT training?
>
>
> As suggested by the reviewer (and reviewer o26f), we investigated the relationship between RPT training time and improvements in corrector iterations. Due to the limited time available for this rebuttal, we were unable to complete a full training cycle. Instead, we fine-tuned a model using 200B tokens and the results are presented in the table below. Our findings indicate that increasing training steps does not enhance the model’s correction capabilities. We also fine-tuned the model with fewer tokens, observing that while 30B tokens might be insufficient for corrector sampling, training with 60B tokens yields improvements similar to those reported in the paper.
>
>
> | # Tokens  | Iterations | HumanEval+ | MBPP | GSM8K |
> |---|---|---|---|---|
> | 30B | 0 | 26.8 | 36.8 | 34.2 |
> |  | 1 | 27.4 | 37.2 | 35.9 |
> |  | 2 | 27.4 | 37.0 | 35.6 |
> | 60B | 0 | 28.0 | 38.2 | 35.5 |
> |  | 1 | 29.2 | 39.4 | 35.9 |
> |  | 2 | 28.7 | 39.2 | 36.2 |
> | 100B (paper) | 0 | 27.4 | 39.6 | 35.5 |
> |  | 1 | 27.4 | 40.6 | 37.5 |
> |  | 2 | 28.0 | 39.0 | 36.3 |
> | 200B | 0 | 28.7 | 39.2 | 35.5 |
> |  | 1 | 30.5 | 39.0 | 36.3 |
> |  | 2 | 29.9 | 40.6 | 37.5 |
>
> > In general, how long will it take in RPT inference compared to NTP for window size=2?
>
> Each RPT iteration adds $w$ (window size) additional forwards. We will make this clearer in the revised paper.

---

> > ### Comment · Reviewer_6ySe · 2025-08-04
> >
> > Thanks for your response and new experiment results. My main concern on the training-inference gap is addressed.

---

> > > ### Author Response · Authors · 2025-08-04
> > >
> > > We thank the reviewer for acknowledging that their main concern has been addressed. Given that this was a key point in the initial (negative) assessment, we would be grateful if the reviewer would be willing to raise their score to reflect this change.

---

> > > > ### Comment · Reviewer_6ySe · 2025-08-05
> > > >
> > > > Sure, I have updated my score to 4.

---

### Official Review · Reviewer_wdtD · 2025-07-02

**Clarity:** 2
**Significance:** 4
**Originality:** 3
**Rating:** 5
**Confidence:** 4

**Summary:**

Corrector sampling proposes Resample Previous Tokens (RPT): a novel language modelling task and sampling strategy that facilitates revision of previously generated tokens within a fixed sized lookback window conditioned both on the prior context and subsequently generated tokens. The authors formally introduce their method and provide a robust theoretical analysis of RPT and Next Token Prediction (NTP) asymptotic errors. The error analysis is empirically validated with a synthetic experiment which agrees with their analysis that RPT exhibits a lower error than NTP. Finally, the authors pre-train a Llama-3-8B model from scratch for 900B tokens with the classic NTP task and use this checkpoint to further fine-tune two variants: 1) AR-F which is fine-tuned on an additional 100B tokens in the pure NTP task; and, 2) a model trained for RPT by randomly swapping target indices and carefully manipulating positional encoding. The results demonstrate that even in the purely auto-regressive sampling setting, RPT improves model quality on a variety of reasoning and coding benchmarks. Further, when using iterative RPT sampling, the model quality is further improved.

**Questions:**

## Key questions
* Could the authors speculate on the cause of surprising results in Table 1. I would have naively expected AR-F to outperform AR-C in all cases and similarly was surprised to see AR-C significantly outperform in the PHP category.
* Was only pre-training data used? Any SFT or RL training? Please discuss the high-level content of the corpus used; was it curated for the benchmark tasks specifically or drawn from a larger, more general dataset?
* Can RPT training be extended to previously instruct-tuned checkpoints? What are some potential challenges to extending RPT training to this paradigm?
* What is the computational overhead of RPT samping? Am I correct in my understanding that for a single RPT iteration (k=1) and a window size w, each output token requires w forward passes?
* Please clarify the imprecise or misunderstood notation as requested in the weaknesses section.

## Minor/Typos:
* L129: “next-token-**rediction** (**NPT**)”
* L132: NPT used in lieu of NTP twice.
* L193: Should be revised as “left” and “right” instead of “top” and “bottom”, respectively.
* L304: “**fo**”

**Ethical Concerns:**

["NO or VERY MINOR ethics concerns only"]

**Final Justification:**

The majority of my concerns were addressed during rebuttal. The AR-C variant outperforming AR-F remains a somewhat surprising result; however, the author's explanation regarding some variability in accuracy during AR training is sufficiently explanatory. The comparison between BoN sampling and corrector sampling suggests that corrector sampling offers a net benefit even in compute matched settings, further convincing me that the proposed method is a significant contribution worthy of inclusion in the proceedings.

**Limitations:**

* **No**. The authors stated in the checklist that limitations were discussed; however, I could not find any related discussion. I encourage the authors to consider the limitations of their method in particular from the perspectives of computational efficiency, performance (latency/throughput), and extending RPT training to previously instruct-tuned models.

**Quality:**

4

**Strengths And Weaknesses:**

## Strengths
* The proposed methodology appears to be novel.
* The submission includes both high-quality theoretical analysis and compelling empirical evidence that the proposed technique materially benefits the RPT trained models compared to pure NTP models.
* The submission is relatively clear; however, some notation could be made more consistent and a few statements clarified as outlined below.
* The results are very promising and suggest a potential improvement to current the pre-training paradigm. In particular, it is very significant that RPT trained models improve in quality *even when auto-regressively sampled* (i.e., with no RPT iterations to actually revise past tokens). This suggests that the representations learned by the model during RPT are beneficially even for NTP. This is a very important outcome since, if I understand the sampling procedure correctly, RPT requires significantly more computation than pure auto-regressive decoding.

## Weaknesses
* The main weaknesses are some unexpected results in the empirical results where AR-C out performs AR-F and/or the RPT variants; for instance, the C# and PHP tasks for AR-C outperforming AR-F and RPT, respectively.
* Some additional information on the empirical experiments would benefit the reader and improve reproducibility.
* While the authors demonstrate the benefit of RPT training using only 10% of the pre-training budget; this still represents a very significant amount of training (100B tokens). It could be particularly impactful if RPT training could be extended to a more lightweight fine-tuning setting of instruct tuned model checkpoints.
* Further discussion on the computational requirements of RPT sampling would improve the work and help clearly establish some of the limitations of RPT sampling.
* Additionally to the above, since RPT requires additional computation, an interesting ablation would be to compare RPT with compute-matched test-time-scaling such as beam search or Best-of-N parallel sampling with the NPT trained checkpoint. Does RPT sampling provide a benefit in a strict compute-matched setting?
* Some potentially imprecise notation:
	* L72: Shouldn’t $w=1$ represent the NTP sampling procedure as defined? The most recent token generated in equation 6 is $x_{i+w-1}$. This seems to be the case in algorithm 1 where w is explicitly restricted to $\geq 2$.
	* L80: It appears that $k$ should be restricted to the domain $k \in \{1, …, n-w+1\}$ since we can’t permute tokens within a window size of the last token in a sequence; there would be no future tokens to condition RPT sampling with.
* In Table 1, $n$ is used to signify the number of RPT iterations. However, elsewhere in the paper such as section 5.3 $k$ is used. Unifying the notation between these sections would improve readability. Further, it’s unclear what n=0.5 or 1.5 refer to; does 0.5 represent a sliding window of 2 with a stride of 2 such that only every second token is resampled?
* In Appendix B.2, the authors refer to “naive” sampling. It seems like this is simply nucleus samping. If so, please simply state nucleus sampling as it is less confusing to the reader than introducing a second, less common term.
* L263 states that “greedy decoding is always used in the main paper”, but based on section B.2 it seems like what was actually reported in table 1 is greedy decoding **+ confidence**. Please clarify.

---

> ### Author Rebuttal · Authors · 2025-07-30
>
> > The main weaknesses are some unexpected results in the empirical results where AR-C out performs AR-F and/or the RPT variants; for instance, the C# and PHP tasks for AR-C outperforming AR-F and RPT, respectively.
>
> > Could the authors speculate on the cause of surprising results in Table 1. I would have naively expected AR-F to outperform AR-C in all cases and similarly was surprised to see AR-C significantly outperform in the PHP category.
>
> We agree this is surprising, but we thought it’s useful to report this nonetheless. Our only explanation is that we notice in general that benchmarks results for AR training are not always monotonic during the pretraining. Consequently, also RPT is not always monotonic. However, note that we do compare AR-F and RPT after 1T token (at the final state of the cosine scheduler).
>
> > Some additional information on the empirical experiments would benefit the reader and improve reproducibility.
>
> If the reviewer can specify where they feel information is missing we will happily elaborate in the text.
>
>
> > While the authors demonstrate the benefit of RPT training using only 10% of the pre-training budget; this still represents a very significant amount of training (100B tokens). It could be particularly impactful if RPT training could be extended to a more lightweight fine-tuning setting of instruct tuned model checkpoints.
>
> Following the reviewer's suggestion, we conducted additional experiments with 30B and 60B tokens. The results, presented in the table below, indicate that 30B tokens are insufficient to fully unlock RPT's capabilities, while 60B tokens appear to be adequate.
>
> | # Tokens  | Iterations | HumanEval+ | MBPP | GSM8K |
> |---|---|---|---|---|
> | 30B | 0 | 26.8 | 36.8 | 34.2 |
> |  | 1 | 27.4 | 37.2 | 35.9 |
> |  | 2 | 27.4 | 37.0 | 35.6 |
> | 60B | 0 | 28.0 | 38.2 | 35.5 |
> |  | 1 | 29.2 | 39.4 | 35.9 |
> |  | 2 | 28.7 | 39.2 | 36.2 |
> | 100B (paper) | 0 | 27.4 | 39.6 | 35.5 |
> |  | 1 | 27.4 | 40.6 | 37.5 |
> |  | 2 | 28.0 | 39.0 | 36.3 |
> | 200B | 0 | 28.7 | 39.2 | 35.5 |
> |  | 1 | 30.5 | 39.0 | 36.3 |
> |  | 2 | 29.9 | 40.6 | 37.5 |
>
> > Further discussion on the computational requirements of RPT sampling would improve the work and help clearly establish some of the limitations of RPT sampling.
>
> > What is the computational overhead of RPT sampling? Am I correct in my understanding that for a single RPT iteration ($k=1$) and a window size w, each output token requires w forward passes?
>
> Correct. We will add the following to the revised paper (and mention it as a limitation). Each RPT iteration adds $w$ (window size) additional forwards.
>
>
> > Additionally to the above, since RPT requires additional computation, an interesting ablation would be to compare RPT with compute-matched test-time-scaling such as beam search or Best-of-N parallel sampling with the NPT trained checkpoint. Does RPT sampling provide a benefit in a strict compute-matched setting?
>
> As suggested by the reviewer, we incorporated best-of-n results. In a non-verifiable setup (as we report in the paper), we generate $n$ samples and select the one with the highest model-assigned likelihood. While this baseline does improve benchmark performance, corrector iterations consistently yield better results with more significant improvements.
>
> | Best of n | HumanEval+ | MBPP | GSM8K |   |
> |-----------|------------|------|-------|---|
> | 1         | 25.6       | 39.0 | 35.2  |   |
> | 2         | 25.6       | 38.6 | 35.5  |   |
> | 3         | 26.2       | 39.5 | 33.3  |   |
> | 4         | 26.2       | 40.6 | 36.3  |   |
>
> > Some potentially imprecise notation:  1) L72: Shouldn’t $w=1$ represent the NTP sampling procedure as defined? The most recent token generated in equation 6 is . This seems to be the case in algorithm 1 where w is explicitly restricted to >= 2) L80: It appears that should be restricted to the domain $k\in \\{1,..,n-w+1\\}$ since we can’t permute tokens within a window size of the last token in a sequence; there would be no future tokens to condition RPT sampling with.
>
> Thanks for pointing it out, we will fix it!
>
> > In Table 1, $n$ is used to signify the number of RPT iterations. However, elsewhere in the paper such as section 5.3 $k$ is used. Unifying the notation between these sections would improve readability. Further, it’s unclear what $n=0.5$ or $1.5$ refer to; does $0.5$ represent a sliding window of $2$ with a stride of $2$ such that only every second token is resampled?
>
> **Typo of $n$ and $k$**: Thanks for pointing out, we will fix it in the revised manuscript.
>
> **Fractional iterations**: Fractional values of $n$ refer to iterations that stop at the first token. For example, for $n=1.5$, we have:
> * $x_i \sim \hat{p}(x_i | x_{< i})$
> * $x_{i+1} \sim \hat{p}(x_{i+1} | x_{< i+1})$
> * $x_{i} \sim \hat{p}(x_i | x_{< i}, x_{i+1})$
>
> We will add this clarification to the manuscript.
>
> > In Appendix B.2, the authors refer to “naive” sampling. It seems like this is simply nucleus sampling. If so, please simply state nucleus sampling as it is less confusing to the reader than introducing a second, less common term.
>
> Thanks, we will fix it.
>
> > L263 states that “greedy decoding is always used in the main paper”, but based on section B.2 it seems like what was actually reported in table 1 is greedy decoding + confidence. Please clarify.
>
> All main Table results are with greedy sampling and confidence. We will clarify in the manuscript.
>
> > Was only pre-training data used? Any SFT or RL training? Please discuss the high-level content of the corpus used; was it curated for the benchmark tasks specifically or drawn from a larger, more general dataset?
>
> Throughout the paper, we used a fixed pretraining web-scale datamix (for reference, similar to DCLM [1]). We did not use any high-quality curated data such as being used in SFT or RL training.
>
> > Can RPT training be extended to previously instruct-tuned checkpoints? What are some potential challenges to extending RPT training to this paradigm?
>
> RPT adoption for such models should be straightforward. One thing to note is that, similarly to AR SFT training, we should take RPT loss only on certain parts of the data sequences.
>
> > Please clarify the imprecise or misunderstood notation as requested in the weaknesses section.
>
> > Typos: L129: “next-token-rediction (NPT)” L132: NPT used in lieu of NTP twice. L193: Should be revised as “left” and “right” instead of “top” and “bottom”, respectively. L304: “fo”
>
> Thanks for pointing out, we will fix these in the revised manuscript.
>
> [1] DataComp-LM: In search of the next generation of training sets for language models, Li et al., 2024.

---

> > ### Comment · Reviewer_wdtD · 2025-08-03
> >
> > I thank the authors for the detailed rebuttal. The majority of my concerns have been addressed and I believe the revisions noted above will improve the manuscript. I will maintain my original score.
> >
> > > If the reviewer can specify where they feel information is missing we will happily elaborate in the text.
> >
> > The information regarding the pretraining dataset has satisfied my original concerns regarding reproducibility.
> >
> > The comparison between BoN sampling and corrector sampling suggests that corrector sampling offers a net benefit even in compute matched settings. This may be worth highlighting in the camera-ready version in my opinion.

---

### Official Review · Reviewer_o26f · 2025-07-06

**Clarity:** 3
**Significance:** 2
**Originality:** 3
**Rating:** 4
**Confidence:** 3

**Summary:**

The paper introduces Resample-Previous-Tokens (RPT) for autoregressive (AR) LLMs that aims to mitigate the issue of error accumulation during token generation. Traditional AR models suffer from irrevocable errors in token generation, as once a token is produced, it cannot be revised. RPT iteratively revisits previously generated tokens within a fixed-size window and replacing them based on context. This method is designed to be easily integrated into existing AR models. The paper demonstrates that fine-tuning a pretrained 8B parameter model with RPT results in a 5-10% improvement in reasoning and coding tasks.

**Questions:**

1. What is the ratio of NTP and RPT? How did the ratio of RPT have the effect on the model performance?

2. The window size w is set to 3 during training while w=2 during sampling, why did not keep them the same?

3. The RPT shares the similar format to FIM (Fill-in-the-Middle). The discussion of relation to FIM could be helpful.

4. In Table 1, what is the meaning and choices of $n$? A further discussion and analysis is required.

5. The paper continue-trained LLM on RPT for 100 tokens after AR training for 900B tokens. How is the scaling trend on pure RPT?

6. How about mixing the NTP and RPT? What is the performance of it?

7. How did the author conduct RPT during inference on the evaluated experiments? A comparison results among varying sampling hyperparameters such as $w$ is required.

7. What is the weakness of RPT brought during pretraining?

**Ethical Concerns:**

["NO or VERY MINOR ethics concerns only"]

**Final Justification:**

The author states many misuse of hyperparameters in the paper, which requires modification in the next version. Also, it adds additional ablation study to demonstrate the hyperparameter choice. Overall, this work could provide insight to the LLM society, I am increasing my score.

**Limitations:**

yes

**Quality:**

3

**Strengths And Weaknesses:**

**Pros**:

1. The paper addresses a significant issue in AR: error accumulation due to the fixed left-to-right generation process. RPT provides a novel and practical solution to enhance the model performance.

2. RPT is straightforward and easy to implement in AR training. It adds minimal computational overhead.

3. The paper provides empirical validation of RPT via reasoning and coding tasks, e.g., HumanEval+, GSM8K, MBPP.

4. The authors give a solid theoretical foundation for the method, which provides an asymptotic error analysis comparing RPT with NTP sampling.

**Cons**:

1. The paper does not provide sufficient detail on how RPT training influences the model during inference.

2. In Table 1, the meaning of $n=1.5, 2.5$ is not clearly explained. Further discussion is needed to clarify this term and provide more context. Does it refer to different values of the window size or some other hyperparameter? Also, the results would benefit from a comprehensive analysis of how varying $n$ impacts the model’s performance.

3. It lacks comprehensive analysis on the introduced extra hyperparameters, such as $s$, $q$, $n$, $w$. The choices and selection strategy are required to be carefully compared.

4. RPT introduces extra relative position embedding layers, which requires further discussion about its potential drawbacks. It introduces extra parameters to optimize. How are these embeddings injected into the model’s layer inputs (added on each layer or on the token embedding layer), and what are the implications of this additional layer in terms of optimization and model efficiency?

---

> ### Author Rebuttal · Authors · 2025-07-30
>
> > In Table 1, the meaning of $n=1.5,2.5$ is not clearly explained. Further discussion is needed to clarify this term and provide more context. Does it refer to different values of the window size or some other hyperparameter? Also, the results would benefit from a comprehensive analysis of how varying n impacts the model’s performance.
>
> > It lacks comprehensive analysis on the introduced extra hyperparameters, such as $s,q,n,w$. The choices and selection strategy are required to be carefully compared
>
> > How did the author conduct RPT during inference on the evaluated experiments? A comparison results among varying sampling hyperparameters such as $w$ is required
>
> * Fractional values of $n$ (we had this mistakenly called also $k$, will fix) refer to iterations that stop at the first token. For example, for $n=1.5$, we have:
>     * $x_i \sim \hat{p}(x_i | x_{< i})$
>     * $x_{i+1} \sim \hat{p}(x_{i+1} | x_{< i+1})$
>     * $x_{i} \sim \hat{p}(x_i | x_{< i}, x_{i+1})$
> * We will add this clarification to the manuscript.
> * Table 1 and the tables in Appendix B.1 and B.2 show results for $n\in \\{0,1,2\\}$ under various window sizes $w$ and greedy decoding and confidence.
> * Following the reviewer's suggestion, for the rebuttal we added table A and B below, showing the effect of the permutation probability $s$, and $q$ - the probability of moving a token forward within a permuted sequence (see Section 2.1). As can be seen in these tables our method is robust to these hyperparameters. We will add these results to the revised manuscript.
>
> **Table A**
>
> Fixing $q=0.02$ and
>
> | s           | iterations | HumanEval+ | MBPP | GSM8K |
> |-------------|------------|------------|------|-------|
> | 0.1         | 0          | 28.0       | 38.0 | 35.4  |
> |             | 1          | 30.5       | 38.8 | 36.2  |
> |             | 2          | 29.3       | 38.6 | 35.9  |
> | 0.5 (paper) | 0          | 27.4       | 39.6 | 35.5  |
> |             | 1          | 27.4       | 40.6 | 37.5  |
> |             | 2          | 28.0       | 39.0 | 36.3  |
> | 1.0         | 0          | 26.8       | 36.8 | 34.2  |
> |             | 1          | 28.0       | 39.0 | 34.8  |
> |             | 2          | 28.7       | 39.4 | 35.6  |
>
>
> **Table B**
>
> Fixing $s=0.5$ and
>
>
> | q            | iterations | HumanEval+ | MBPP | GSM8K |
> |--------------|------------|------------|------|-------|
> | 0.01         | 0          | 26.8       | 38.2 | 34.0  |
> |              | 1          | 27.4       | 38.8 | 36.2  |
> |              | 2          | 28.6       | 40.0 | 35.9  |
> | 0.02 (paper) | 0          | 27.4       | 39.6 | 35.5  |
> |              | 1          | 27.4       | 40.6 | 37.5  |
> |              | 2          | 28.0       | 39.0 | 36.3  |
> | 0.05         | 0          | 25.6       | 38.0 | 34.2  |
> |              | 1          | 26.8       | 39.3 | 36.2  |
> |              | 2          | 26.2       | 38.3 | 34.3  |
> | 0.1          | 0          | 25.6       | 38.2 | 34.9  |
> |              | 1          | 26.2       | 38.6 | 35.6  |
> |              | 2          | 25.6       | 38.4 | 35.9  |
>
> > What is the ratio of NTP and RPT? How did the ratio of RPT have the effect on the model performance?
>
> In practice, the ratio of NTP and RPT is fixed by $s$ and $q$ corresponding to the sequence permutation probability and token forward push probability. We add in Tables A and B their effect on the model performance.
>
> > RPT introduces extra relative position embedding layers, which requires further discussion about its potential drawbacks. It introduces extra parameters to optimize. How are these embeddings injected into the model’s layer inputs (added on each layer or on the token embedding layer), and what are the implications of this additional layer in terms of optimization and model efficiency?
>
> The relative position is added only to the token embedding layer, and introduces a very small additional parameter count: 12k parameters for relative positions of $\\{-1,0,1\\}$ where $w=2$ and an 8B model, versus 500m parameters used by the token embedding layer. In optimization, only $O(w)$ new vectors are added to the embedding layers, which is negligible compared to vocabulary size . During inference, this additional lookup table computation is parallelized with the token embedding layer and does not introduce any overhead in practice. We will add this discussion to the revised manuscript.
>
>
> > The window size w is set to 3 during training while $w=2$ during sampling, why did not keep them the same?
>
> Sampling with $w=3$ (Appendix B.1) achieved comparable results to $w=2$ (Table 1, main paper) however $w=2$ is a faster sampling method, so generally preferable. We do believe that it is quite possible that in other benchmarks $w\geq 3$ could improve over $w=2$, but for the benchmarks we tested in this paper that was not the case.
>
>
> > The RPT shares the similar format to FIM (Fill-in-the-Middle). The discussion of relation to FIM could be helpful.
>
>
> The fill-in-the-middle method decomposes the sequence to prefix, middle, and suffix and then orders it in PSM (prefix, suffix, middle) or SPM (suffix, prefix, middle) fashion. During generation, the model is required to predict the ‘middle’ part, which can introduce completions of arbitrary length with known issues of when to terminate the middle part. Our corrector iterations do not suffer from this issue as they are trained on fixed token lengths. Another question is if arbitrary middle completion has theoretical error correcting properties as RPT, a good question for future work. We will add this discussion to the related work section.
>
>
> > In Table 1, what is the meaning and choices of $n$? A further discussion and analysis is required.
>
> The choices of $n$ in Table 1, refer to the actual number of corrector iterations we employ. Please note that we have a typo where we used $n$ and $k$ with the same meaning, we will fix it in the revised paper.
>
> > The paper continue-trained LLM on RPT for 100 tokens after AR training for 900B tokens. How is the scaling trend on pure RPT?
>
> As suggested by the reviewer (and reviewer 6ySe), we investigated the relationship between RPT training time and improvements in corrector iterations. Due to the limited time available for this rebuttal, we were unable to complete a full training cycle. Instead, we fine-tuned a model using 200B tokens and the results are presented in the table below. Our findings indicate that increasing training steps does not enhance the model’s correction capabilities. We also fine-tuned the model with fewer tokens, observing that while 30B tokens might be insufficient for corrector sampling, training with 60B tokens yields improvements similar to those reported in the paper.
>
>
> | # Tokens  | Iterations | HumanEval+ | MBPP | GSM8K |
> |---|---|---|---|---|
> | 30B | 0 | 26.8 | 36.8 | 34.2 |
> |  | 1 | 27.4 | 37.2 | 35.9 |
> |  | 2 | 27.4 | 37.0 | 35.6 |
> | 60B | 0 | 28.0 | 38.2 | 35.5 |
> |  | 1 | 29.2 | 39.4 | 35.9 |
> |  | 2 | 28.7 | 39.2 | 36.2 |
> | 100B (paper) | 0 | 27.4 | 39.6 | 35.5 |
> |  | 1 | 27.4 | 40.6 | 37.5 |
> |  | 2 | 28.0 | 39.0 | 36.3 |
> | 200B | 0 | 28.7 | 39.2 | 35.5 |
> |  | 1 | 30.5 | 39.0 | 36.3 |
> |  | 2 | 29.9 | 40.6 | 37.5 |
>
> > How about mixing the NTP and RPT? What is the performance of it?
>
> Not sure we fully understand the reviewer question here. Our training indeed mixes RPT and NTP (see Section 2.1, paragraph “Training”). The probability of permutation $s$ sets the ratio between native NTP training and RPT training; see also Table A in the general response for its effect on model performance. If the reviewer means something else, we would be happy to clarify further.
>
> > What is the weakness of RPT brought during pretraining?
>
>
> During pretraining, we add an additional relative position embedding layer (discussed above), however an efficient implementation of this layer does not introduce any overhead. On the flip side, please note that pretraining with RPT improves the results compared to only NTP training, see Table 1 $n=1$ (NTP sampling after RPT training) versus AR-F. We attribute this to the nature of our method, which introduces a non-trival data/task augmentation.
>
> > The paper does not provide sufficient detail on how RPT training influences the model during inference.
>
> We are not sure what influence the reviewer is referring to. As Table 1 shows (see, $n=0$ row), sampling purely in AR fashion after RPT training seems to somewhat improve results over training with AR and sampling with AR. If the reviewer has something else in mind, we would be happy to clarify further.

---

> > ### Author Response · Authors · 2025-08-06
> >
> > As discussion period coming to an end - we would like to draw the reviewer's attention to the comprehensive rebuttal and new results we produced to address their comments and would greatly appreciate a response. Thanks!

---

> ### Comment · Reviewer_o26f · 2025-08-09
>
> Thank you for providing the detailed response and additional experimental results for rebuttal. The response have addressed most of my concerns. Accordingly, I am increasing the overall assessment. Further follow-up questions are:
>
> 1. What if applying RPT to pure NTP training, with the extra embeddings completely removed? This could simplify the process and make it easier to generalize to common transformer architectures.
>
> 2. The paper introduces may additional hyperparameters such as $s$, $q$, $n$, ... (in. Alg.1, another $m$). Could the authors clearly explain the exact meaning (and difference) of each of them? It can pose heavy hyperparam-tuning costs on pretraining. All essential for the model's performance?
>
> 3. In Table A/B, how was the evaluation conducted? Does “iterations > 0” indicate that iterative RPT sampling was used for each generated token? If so, how about the additional benefits gained from the extra inference compute (c.f. test-time scaling)?

---

> ### Author Response · Authors · 2025-08-09
>
> We thank the reviewer for raising their score. Below we address the remaining concerns.
>
> > What if applying RPT to pure NTP training, with the extra embeddings completely removed? This could simplify the process and make it easier to generalize to common transformer architectures.
>
> We don’t see a way to incorporate RPT without any additional information provided to the network. Let us give an example: Assume the model needs to predict next token to $p( x | \text{“The”, “cat”} )$ but not given any extra information - it will likely predict a reasonable subsequent token such as “sat”. However, if this is the RPT sampling stage we would expect a prediction of a middle token such as “fat”. Without solving this ambiguity we don’t see how the model can distinguish between the cases.
> One option to avoid adding the extra positional embedding layer is to consider introducing new tokens (which implicitly also add more parameters through the existing embedding layer).
>
> > The paper introduces may additional hyperparameters such as $s,q,n,..$ (in Alg. 1, another $m$). Could the authors clearly explain the exact meaning (and difference) of each of them? It can pose heavy hyperparam-tuning costs on pretraining. All essential for the model's performance?
>
> Here is the explanation of the meaning and difference of the hyper-parameters:
> $s$ - for each sequence, the probability of being permuted (described in detail in lines 90-91).
> $q$ - for each token in the sequence, the probability of moving this token $w-1$ places forward (described in detail in lines 92-97).
> $w$ - window size.
> $n$ ($k$ after typo fix) - RPT iterations; in the rebuttal time we realized that we had typo and used $n$ instead of $k$ in some parts of the submission. We will fix it in the camera ready.
> $m$ - is not a part of the method, it is the number of training iterations, we will clarify.
>
> We argue that the method is fairly robust to the hyper-param choices. In our above response to the reviewer we have added ablations on the $s,q$ parameters (see Tables A and B above) demonstrating this robustness. The $n$ ($k$ after typo fix) parameter influence is shown in Table 1 and Section B in Appendix.
>
> > In Table A/B, how was the evaluation conducted? Does “iterations > 0” indicate that iterative RPT sampling was used for each generated token? If so, how about the additional benefits gained from the extra inference compute (c.f. test-time scaling)?
>
> Yes, "iterations > 0” means RPT sampling was employed for each token. The performance gain from the extra inference compute is a 5-10% improvement in benchmark performance, similar to the improvements observed in the submission. If we misunderstood the reviewer’s question, we are happy to clarify further.

---

> ### Comment · Reviewer_o26f · 2025-08-09
>
> Thanks for the response.
>
> 1. Regarding removing learned position encoding: It would be more consistent if we can introduce extra special token as RPT indicators, as FIM did.
>
> 2. For extra inference compute, I’m happy to know more about the relation between extra inference cost versus performance gain by RPT sampling.
>
> Based on above discussions, this work could provide novel insights to pre-training practitioners. I’m also happy to know whether the authors would open source their code afterwards.
>
>
> Based on above response, I’m increasing my overall score.

---

### Note · Authors · 2025-08-15

We sincerely thank all the reviewers for their time and insightful feedback. We appreciate the recognition of our method's novelty and strong results. We would also like to thank Reviewers o26f and 6ySe for raising their scores and recommending acceptance, resulting in all reviewers recommending for acceptance.

In response to Reviewer o26f's final question (which we didn’t get a chance to respond due to the system closing soon after it was posted), we plan to make available the main code.

To address the points raised, we conducted several new experiments, which we will integrate into the final paper:

* Hyperparameter Sensitivity Analysis: We found that our method is robust to the choice of parameters $s$ and $q$, and does not require extensive or sensitive hyperparameter tuning to be effective.
* Training Data Ablation: We conclude that our method is training-efficient. While 30B tokens are insufficient, it achieves significant gains with just 60B tokens, and further training does not yield substantial additional improvements.
* Generalization to New Benchmarks: We added results on the TQA and NQ benchmarks, showing that RPT generalizes well beyond coding and math to improve performance on standard QA tasks.

Alongside these new empirical results, we will incorporate all requested clarifications and typo fixes into the camera-ready version.

---

### Decision · Program_Chairs · 2025-09-17

**Decision:**

Accept (poster)

**Comment:**

(a) Summary: The paper introduces Resample-Previous-Tokens (RPT), a novel training and sampling method for autoregressive language models that mitigates error accumulation by allowing the model to revise previously generated tokens within a fixed-size window. Unlike standard next-token prediction, RPT conditions revisions on both past and future context, making prediction easier and reducing asymptotic errors. The authors provide theoretical analysis, synthetic validation, and large-scale experiments, showing that fine-tuning an 8B Llama model with RPT yields 5–10% gains on reasoning and coding benchmarks, while maintaining efficiency and compatibility with existing AR models.

(b) Strengths:
1. The paper proposes a promising solution RPT to tackle the major challenge of error accumulation in AR models caused by their fixed left-to-right generation, while maintaining the next-token-prediction quality.

2. The submission includes both high-quality theoretical analysis and compelling empirical evidence.

3. The paper is well-written and reviewers all agree that the idea is novel.

(c) Weaknesses (after rebuttal):
1. The matter of hyper-parameter choice should be clearly stated in the paper and tested in the ablation study.

2. The effect of this method when there are many tokens, such as 512 tokens, to be modified has not been verified.

(d) Why this decision:
Reviewers all agree that this is a good paper which proposes a promising method to challenge traditional next-token prediction. All reviewers vote consistently positive. AC would follow reviewers' recommendation and recommend for accept.

(e) Summary of discussions:
The rebuttal has done a good job to mitigate some major concerns in the first round of review such as training of the method. However, some others are not addressed, e.g., the effect of this method when there are many tokens, such as 512 tokens, to be modified has not been verified. Given these, AC would recommend Accept (poster) but not higher ranking.